# DISENTANGLING 3D PROTOTYPICAL NETWORKS FOR FEW-SHOT CONCEPT LEARNING

**Mihir Prabhudesai**[*1]**, Shamit Lal**[*1]**, Darshan Patil**[*†2]**, Hsiao-Yu Tung**[1]**, Adam W Harley**[1]**,
Katerina Fragkiadaki**[1]
[1]Carnegie Mellon University    [2]Mila, University of Montreal
{mprabhud,shamitl}@cs.cmu.edu,darshan.patil@mila.quebec,
{htung, aharley, katef}@cs.cmu.edu

## ABSTRACT

We present neural architectures that disentangle RGB-D images into objects' shapes and styles and a map of the background scene, and explore their applications for few-shot 3D object detection and few-shot concept classification. Our networks incorporate architectural biases that reflect the image formation process, 3D geometry of the world scene, and shape-style interplay. They are trained end-to-end self-supervised by predicting views in static scenes, alongside a small number of 3D object boxes. Objects and scenes are represented in terms of 3D feature grids in the bottleneck of the network. We show that the proposed 3D neural representations are compositional: they can generate novel 3D scene feature maps by mixing object shapes and styles, resizing and adding the resulting object 3D feature maps over background scene feature maps. We show that classifiers for object categories, color, materials, and spatial relationships trained over the disentangled 3D feature sub-spaces generalize better with dramatically fewer examples than the current state-of-the-art, and enable a visual question answering system that uses them as its modules to generalize one-shot to novel objects in the scene.

## 1 INTRODUCTION

Humans can learn new concepts from just one or a few samples. Consider the example in Figure 1. Assuming there is a person who has no prior knowledge about *blue* and *carrot*, by showing this person an image of a blue carrot and telling him "this is an *carrot* with *blue* color", the person can easily generalize from this example to (1) recognizing *carrots* of varying colors, 3D poses and viewing conditions and under novel background scenes, (2) recognizing the color *blue* on different objects, (3) combine these two concepts with other concepts to form a novel object coloring he/she has never seen before, e.g., red carrot or blue tomato and (4) using the newly learned concepts to answer questions regarding the visual scene. Motivated by this, we explore computational models that can achieve these four types of generalization for visual concept learning.

We propose disentangling 3D prototypical networks (D3DP-Nets), a model that learns to disentangle RGB-D images into objects, their 3D locations, sizes, 3D shapes and styles, and the background scene, as shown in Figure 2. Our model can learn to detect objects from a few 3D object bounding box annotations and can further disentangle objects into different attributes through a self-supervised view prediction task. Specifically, D3DP-Nets uses differentiable unprojection and rendering operations to go back and forth between the input RGB-D (2.5D) image and a 3D scene feature map. From the scene feature map, our model learns to detect objects and disentangles each object into a 3D shape code and an 1D style code through a shape/style disentangling antoencoder. We use adaptive instance normalization layers (Huang & Belongie, 2017) to encourage shape/style disentanglement within each object. Our key intuition is to represent objects and their shapes in terms of **3D feature representations disentangled from style variability** so that the model can correspond objects with similar shape by explicitly rotating and scaling their 3D shape representations during matching.

Project page: https://mihirp1998.github.io/project_pages/d3dp/
[*]Equal contribution
[†]Work done while at Carnegie Mellon University

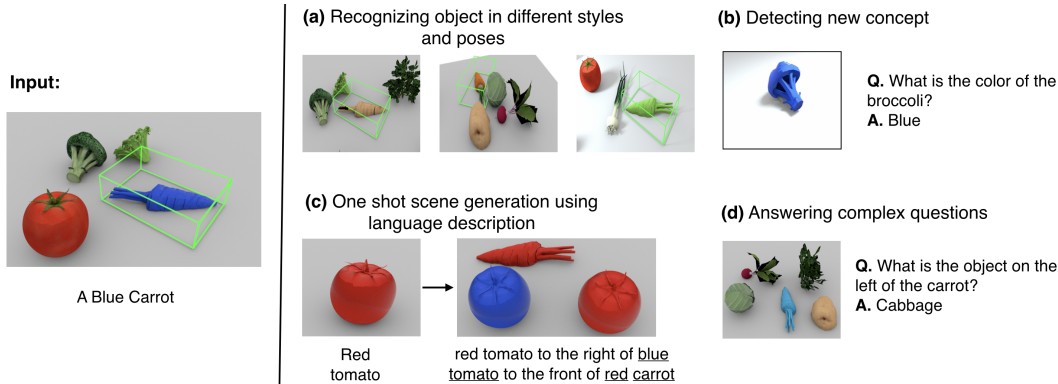

Figure 1: Given a single image-language example regarding new concepts (e.g., blue and carrot), our model can parse the object into its shape and style codes and ground them with *Blue* and *Carrot* labels, respectively. On the right, we show tasks the proposed model can achieve using this grounding.(a) It can detect the object under novel style, novel pose, and in novel scene arrangements and viewpoints. (b) It can detect a new concept like *blue broccoli*. (c) It can imagine scenes with the new concepts. (d) It can answer complex questions about the scene.

With the disentangled representations, D3DP-Nets can recognize new concepts regarding object shapes, styles and spatial arrangements from a few human-supplied labels by training concept classifiers only on the relevant feature subspace. Our model learns object shapes on shape codes, object colors and textures on style codes, and object spatial arrangements on object 3D locations. We show in the supplementary how the features relevant for each linguistic concept can be inferred from a few contrasting examples. Thus the classifiers attend only to the essential property of the concept and ignore irrelevant visual features. This allows them to generalize with far fewer examples and can recognize novel attribute compositions not present in the training data.

We test D3DP-Nets in few-shot concept learning, visual question answering (VQA) and scene generation. We train concept classifiers for object shapes, object colors/materials, and spatial relationships on our inferred disentangled feature spaces, and show they outperform current state-of-the-art (Mao et al., 2019; Hu et al., 2016), which use 2D representations. We show that a VQA modular network that incorporates our concept classifiers shows improved generalization over the state-of-the-art (Mao et al., 2019) with dramatically fewer examples. Last, we empirically show that D3DP-Nets generalize their view predictions to scenes with novel number, category and styles of objects, and compare against state-of-the-art view predictive architectures of Eslami et al. (2018).

The main contribution of this paper is to identify the importance of using disentangled 3D feature representations for few-shot concept learning. We show the disentangled 3D feature representations can be learned using self-supervised view prediction, and they are useful for detecting and classifying language concepts by training them over the relevant only feature subsets. The proposed model outperforms the current state-of-the-art in VQA in the low data regime and the proposed 3D disentangled representation outperforms similar 2D or 2.5D ones in few-shot concept classification.

## 2 RELATION TO PREVIOUS WORKS

**Few-shot concept learning** Few-shot learning methods attempt to learn a new concept from one or a few annotated examples at test time, yet, at training time, these models still require *labelled* datasets which annotate a group of images as "belonging to the same category"(Koch et al., 2015; Vinyals et al., 2016b). Metric-based few-shot learning approaches (Snell et al., 2017; Qi et al., 2018; Schwartz et al., 2018; Vinyals et al., 2016a) aim at learning an embedding space in which objects of the same category are closer in the latent space than objects that belong to different categories. These models needs to be trained with several (annotated) image collections, where each collection contains images of the same object category. Works of Misra et al. (2017); Purushwalkam et al. (2019); Nikolaus et al. (2019); Tokmakov et al. (2019) compose attribute and nouns to detect novel attribute-noun combinations, but their feature extractors need to be pretrained on large annotated image collections, such as Imagenet, or require annotated data with various attribute compositions. The proposed

model is pretrained by predicting views, without the need for annotations regarding object classes or attributes. Our concept classifiers are related to methods that classify concepts by computing distances to prototypes produced by averaging the (2D CNN) features of few labelled examples (Snell et al., 2017; Qi et al., 2018; Schwartz et al., 2018). The work of Prabhudesai et al. (2020) learns 3D prototypes in a self-supervised manner, but they do not disentangle their representation into style and shape codes. We compare against 2D and 3D few shot learning methods and outperform them by a significant margin. The novel feature of our work is that we learn concept prototypes over disentangled 3D shape and 1D style codes as opposed to entangled 3D or 2D CNN features.

**Learning neural scene representation** Our work builds upon recent view-predictive scene representation learning literature (Tung et al., 2019; Sitzmann et al., 2019; Eslami et al., 2016). Our scene encoders and decoders, the view prediction objective, and the 3D neural bottleneck and ego-stabilization of the 3D feature maps is similar to those proposed in geometry-aware neural networks of Tung et al. (2019). Sitzmann et al. (2019) and Eslami et al. (2016) both encode multiview images of a scene and camera poses into a scene representation, in the form of 2D scene feature maps or an implicit function. Sitzmann et al. (2019) only considers single-object scenes and needs to train a separate model for each object class. We compare generalization of our view predictions against Eslami et al. (2016) and show we have dramatically better generalization across number, type and spatial arrangements of objects. Furthermore, the above approaches do not explicitly disentangle style/shape representations of objects. Zhu et al. (2018) focuses on synthesizing natural images of objects with a disentangled 3D representation, but it remains unclear how to use the learnt embeddings to detect object concepts. Different from most inverse graphics networks (Tung et al., 2017; Kanazawa et al., 2018) that aim to reconstruct detailed 3D occupancy of the objects, our model aims to learn feature representations that can detect an object across pose and scale variability, and use them for concept learning. Our shape-style disentanglement uses adaptive instance normalization layers (Huang et al., 2018; Huang & Belongie, 2017) that have been valuable for disentangling shape and style in 2D images. Here, we use them in a 3D latent feature space.

## 3 DISENTANGLING 3D PROTOTYPICAL NETWORKS (D3DP-NETS)

The architecture of D3DP-Nets is illustrated in Figure 2. D3DP-Nets consists of two main components: (a) an image-to-scene encoder-decoder, and (b) an object shape/style disentanglement encoder-decoder. Next, we describe these components in detail.

### 3.1 IMAGE-TO-SCENE ENCODER-DECODER

A 2D-to-3D scene differentiable encoder $\mathrm{E}^{sc}$ maps an input RGB-D image to a 3D feature map $\mathbf{M} \in \mathbb{R}^{w \times h \times d \times c}$ of the scene, where $w, h, d, c$ denote width, height, depth and number of channels, respectively. Every $(x, y, z)$ grid location in the 3D feature map $\mathbf{M}$ holds a $c$-channel feature vector that describes the semantic and geometric properties of a corresponding 3D physical location in the 3D world scene. We output a binary 3D occupancy map $\mathbf{M}^{occ} \in \{0, 1\}^{w \times h \times d}$ from $\mathbf{M}$ using an occupancy decoder $\mathrm{D}^{occ}$. A differentiable neural renderer $\mathrm{D}^{sc}$ neurally renders a 3D feature map $\mathbf{M}$ to a 2D image and a depth map from a specific viewpoint. When the input to D3DP-Nets is a sequence of images as opposed to a single image, each image $I_t$ in the sequence is encoded to a corresponding 3D per frame map $\mathbf{M}_t$, the 3D rotation and translation of the camera with respect to the frame map of the initial frame $I_0$ is computed and the scene map $\mathbf{M}$ is computed by first rotating and translating $\mathbf{M}_t$ to bring it to the same coordinate frame as $\mathbf{M}_0$ and then averaging with the map built thus far. We will assume camera motion is known and given for this cross frame fusion operation.

D3DP-Nets are self-supervised by view prediction, predicting RGB images and occupancy grids for query viewpoints. We assume there is an agent that can move around in static scenes and observes them from multiple viewpoints. The agent is equipped with a depth sensor and knowledge of its egomotion (proprioception) provided by the simulator in simulated environments. We train the scene encoders and decoders jointly for RGB view prediction and occupancy prediction and errors are backpropagated end-to-end to the parameters of the network:

$$\mathcal{L}^{view-pred} = \|\mathrm{D}^{sc}\left(\mathrm{rotate}(\mathbf{M}, v_q)\right) - I_q\|_1 + \log(1 + \exp(-O_q \cdot \mathrm{D}^{occ}((\mathrm{rotate}(\mathbf{M}, v_q)), v_q))), \quad (1)$$

where $I_q$ and $O_q$ are the ground truth RGB image and occupancy map respectively, $v_q$ is the query view, and $\mathrm{rotate}(\mathbf{M}, v_q)$ is a trilinear resampling operation that rotates the content of a 3D feature

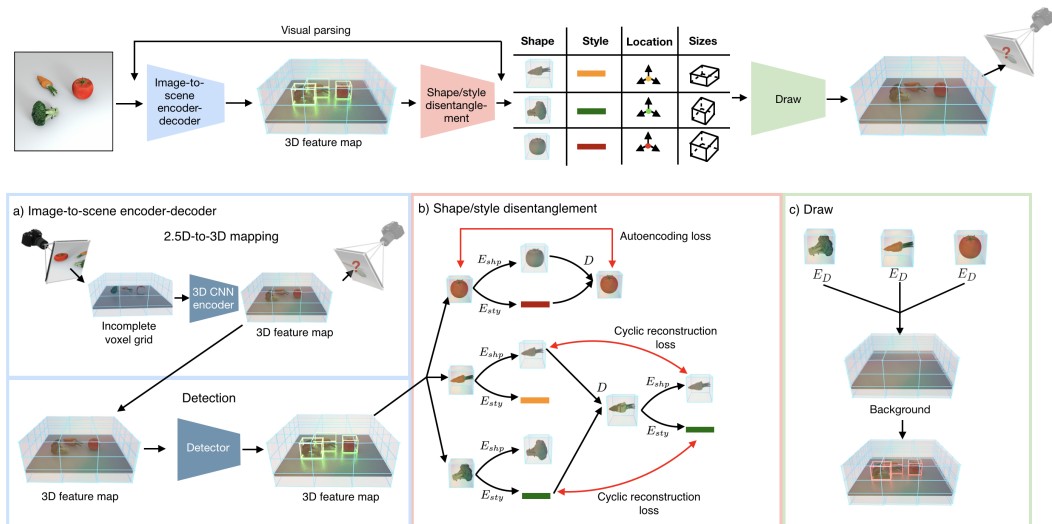

Figure 2: **Architecture for disentangling 3D prototypical networks (D3DP-Nets).** **(a)** Given multi-view posed RGB-D images of scenes as input during training, our model learns to map a single RGB-D image to a completed scene 3D feature map at test time, by training for view prediction. From the completed 3D scene feature map, our model learns to detect objects from the scene. **(b)** In each 3D object box, we apply a shape-style disentanglement autoencoder that disentangles the object-centric feature map to a 3D (feature) shape code and a 1D style code. **(c)** Our model can compose the disentangled representations to generate a novel scene 3D feature map. We urge the readers to refer the video in the supplimentary material for an intuitive understanding of the architecture.

map $\mathbf{M}$ to viewpoint $v_q$. The RGB output is trained with a regression loss, and the occupancy is trained with a logistic classification loss. Occupancy labels are computed through raycasting, similar to Harley et al. (2020). We provide more details on the architecture of our model in the supplementary material. We train a 3D object detector that takes as input the output of the scene feature map $\mathbf{M}$ and predicts 3D axis-aligned bounding boxes, similar to Harley et al. (2020). This is supervised from ground-truth 3D bounding boxes without class labels.

## 3.2 OBJECT SHAPE/STYLE DISENTANGLEMENT

As the style of an image can be understood as a property which is shared across its spatial dimensions, previous works (Huang et al., 2018; Karras et al., 2019) use adaptive instance normalization (Huang & Belongie, 2017) as an inductive bias to do style transfer between a pair of images. D3DP-Nets uses this same inductive bias in its decoder to disentangle the style and 3D shape of an object. We believe that 3D shape is not analogous to 3D occupancy, but it is a blend of 3D occupancy and texture (spatial arrangement of color intensities).

Given a set of 3D object boxes $\{b^o | o = 1 \cdots |\mathcal{O}|\}$ where $\mathcal{O}$ is the set of objects in the scene, D3DP-Nets obtain corresponding object feature maps $\mathbf{M}^o = \mathrm{crop}(\mathbf{M}, b^o)$ by cropping the scene feature map $\mathbf{M}$ using the 3D bounding box coordinates $b^o$. We use ground-truth 3D boxes at training time and detected boxes at test time. Each object feature map is resized to a fixed resolution of $16 \times 16 \times 16$, and fed to an object-centric autoencoder whose encoding modules predict a 4D shape code $z^o_{\mathrm{shp}} = \mathrm{E}_{shp}(\mathbf{M}^o) \in \mathbb{R}^{w \times h \times d \times c}$ and a 1D style code $z^o_{\mathrm{sty}} = \mathrm{E}_{sty}(\mathbf{M}^o) \in \mathbb{R}^c$. A decoder D composes the two using adaptive instance normalization (AIN) layers (Huang & Belongie, 2017) by adjusting the mean and variance of the 4D shape code based on the 1D style code: $AIN(z, \gamma, \beta) = \gamma \left( \frac{z - \mu(z)}{\sigma(z)} \right) + \beta$, where $z$ is obtained by a 3D convolution on $z_{\mathrm{shp}}$, $\mu$ and $\sigma$ are the channel-wise mean and standard deviation of $z$, and $\beta$ and $\gamma$ are extracted using single-layer perceptrons from $z_{\mathrm{sty}}$. The object encoders and decoders are trained with an autoencoding objective and a cycle-consistency objective which ensure that the shape and style code remain consistent after

composing, decoding and encoding again (see Figure 2 (b)):

$$\mathcal{L}^{dis} = \frac{1}{|\mathcal{O}|} \sum_{o=1}^{|\mathcal{O}|} \left( \underbrace{\|\mathbf{M}^o - \mathrm{D}(\mathrm{E}_{shp}(\mathbf{M}^o), \mathrm{E}_{sty}(\mathbf{M}^o))\|_2}_{\text{autoencoding loss}} + \underbrace{\sum_{i \in \mathcal{O} \setminus o} \mathcal{L}^{c-shp}(\mathbf{M}^o, \mathbf{M}^i) + \mathcal{L}^{c-sty}(\mathbf{M}^o, \mathbf{M}^i)}_{\text{cycle-consistency loss}} \right),$$

(2)

where $\mathcal{L}^{c-shp}(\mathbf{M}^o, \mathbf{M}^i) = \|\mathrm{E}_{shp}(\mathbf{M}^o) - \mathrm{E}_{shp}(\mathrm{D}(\mathrm{E}_{shp}(\mathbf{M}^o), \mathrm{E}_{sty}(\mathbf{M}^i)))\|_2$ is the shape consistency loss and $\mathcal{L}^{c-sty}(\mathbf{M}^o, \mathbf{M}^i) = \|\mathrm{E}_{sty}(\mathbf{M}^o) - \mathrm{E}_{sty}(\mathrm{D}(\mathrm{E}_{shp}(\mathbf{M}^i), \mathrm{E}_{sty}(\mathbf{M}^o)))\|_2$ is the style consistency loss.

We further include a view prediction loss on the synthesized scene feature map $\bar{\mathbf{M}}$, which is composed by replacing each object feature map $\mathbf{M}^o$ with its re-synthesized version $\mathrm{D}(z_{\mathrm{shp}}^o, z_{\mathrm{sty}}^o)$, resized to the original object size, as shown in Figure 2(c). The view prediction reads: $\mathcal{L}^{view-pred-synth} = \|\mathrm{D}^{sc}(\mathrm{rotate}(\bar{\mathbf{M}}, v^{t+1})) - I_{t+1}\|_1$. The total unsupervised optimization loss for D3DP-Nets reads:

$$\mathcal{L}^{uns} = \mathcal{L}^{view-pred} + \mathcal{L}^{view-pred-synth} + \mathcal{L}^{dis}.$$

(3)

### 3.3 3D DISENTANGLED PROTOTYPE LEARNING

Given a set of human annotations in the form of labels for object attributes (shape, color, material, size), our model computes prototypes for each concept (e.g. "red" or "sphere") in an attribute, using only the relevant feature embeddings. For example, object category prototypes are learned on top of shape codes, and material and color prototypes are learned on top of style codes. In order to classify a new object example, we compute the nearest neighbors between the inferred shape and style embeddings from the D3DP-Nets with the prototypes in the prototype dictionary, as shown in Figure 3. This non-parametric classification method allows us to detect objects even from a single example, and also improves when more labels are provided by co-training the underlying feature representation space as in Snell et al. (2017).

To compute the distance between an embedding $x$ and a prototype $y$, we define the following rotation-aware distance metric:

$$\langle x, y \rangle_R = \begin{cases} \langle x, y \rangle & \text{if } x, y \text{ are 1D} \\ \max_{r \in \mathcal{R}} \langle \mathrm{Rotate}(x, r), y \rangle & \text{if } x, y \text{ are 4D} \end{cases}$$

(4)

where $\mathrm{Rotate}(x, r)$ explicitly rotates the content in 3D feature map $x$ with angle $r$ through trilinear interpolation. We exhaustively search across rotations $\mathcal{R}$, in a parallel manner, considering increments of $10°$ along the vertical axis. This is specifically shown in the bottom right of Figure 3 while computing the *Filter Shape* function.

Our model initializes the concept prototypes by averaging the feature codes of the labelled instances. We build color and material concept prototypes, e.g., *red* or *rubber*, by passing the style codes through a color fully connected module and a material fully connected module respectively, and then averaging the outputs. For object category prototypes, we use a rotation-aware averaging over the (4D) object shape embeddings, which are produced by a 3D convolutional neural module over shape codes. Specifically, we find the alignment $r$ for each shape embedding that is used to calculate $\langle z_0, z_i \rangle_R$, and average over the aligned embeddings to create the prototype.

When annotations for concepts are provided, we can jointly finetune our prototypes and neural modules (as well as D3DP-Net weights) using a cross entropy loss, whose logits are inner products between neural embeddings and prototypes. Specifically, given $P(o_a = c) = \frac{exp(\langle f_a(z^o), p_c \rangle_R)}{\sum_{d \in \mathcal{C}_a} exp(\langle f_a(z^o), p_d \rangle_R)}$ where $\langle \cdot, \cdot \rangle_R$ represents the rotation-aware distance metric, $f_a$ is the neural module for attribute $a$, $\mathcal{C}_a$ is the set of concepts for attribute $a$, and $o_a$ is the value of attribute $a$ for object $o$, and $p_c$ is the prototype for concept $c$. The loss used to train prototypes is:

$$\mathcal{L}^{prototype} = -\frac{1}{|\mathcal{O}|} \sum_{o \in \mathcal{O}} \sum_{a \in \mathcal{A}} \sum_{c \in \mathcal{C}_a} \mathbb{1}_{o_a = c} \log P(o_a = c) + \mathbb{1}_{o_a \neq c} \log P(o_a \neq c)$$

(5)

where $\mathcal{A}$ is the set of attributes.

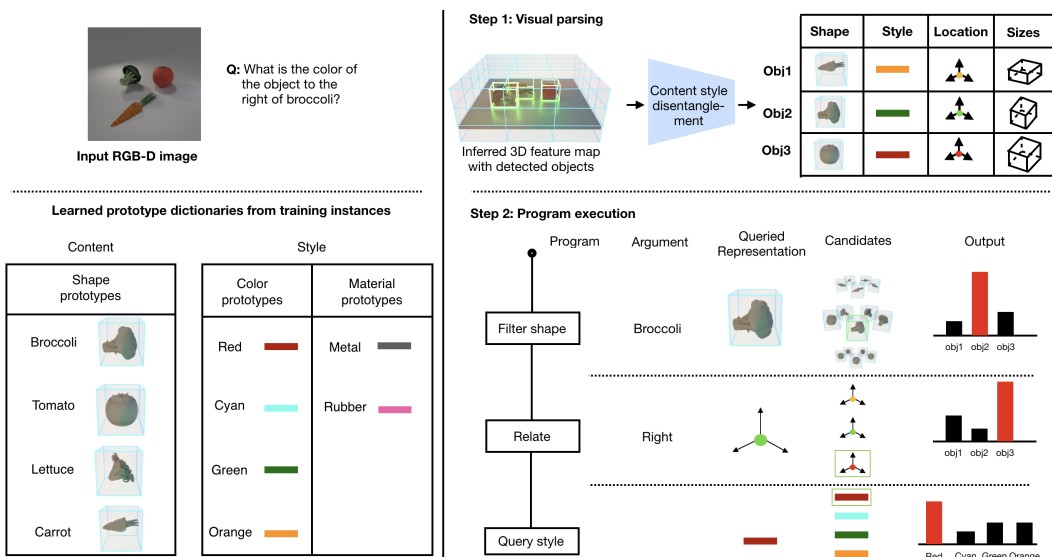

Figure 3: **D3DP-VQA Modular Networks.** Given a question-image pair and a list of learned prototype dictionaries (left), D3DP-Nets parse the visual scene to object shapes, styles, locations and sizes codes (top-right), while the semantic language parser converts the question to an executable program. The generated program is executed sequentially to answer the question (bottom-right). Note that in order to associate different poses of the same shape (Filter Shape), our model does a rotation-aware search between the indexed prototype and the candidate objects.

## 4    EXPERIMENTS

We test D3DP-Nets in few-shot learning of object category, color and material, and compare against state-of-the-art 2D and 2.5D shape-style disentangled CNN representations (Sections 4.1). We integrate these concept classifiers in a visual question answering modular system (see Figure 3) and show it can answer questions about images more accurately than the state-of-the-art in the few-shot regime (Section 4.2). In addition, we test D3DP-Nets on novel 3D scene generation. We also test D3DP-Nets on view prediction and compare against alternative scene representation learning methods (Section 4.3). Furthermore, we show our model can generate a 3D scene (and its 2D image renders) based on a language utterance description (Section 4.4).

### 4.1    FEW-SHOT OBJECT SHAPE AND STYLE CATEGORY LEARNING

We evaluate D3DP-Nets in its ability to classify shape and style concepts from few annotated examples on three datasets: i) CLEVR dataset (Johnson et al., 2017): it is comprised of cubes, spheres and cylinders of various sizes, colors and materials. We consider every unique combination of color and material categories as a single style category. The dataset has 16 style classes and 3 shape classes in total. ii) Real Veggie dataset: it is a real-world scene dataset we collected that contains 800 RGB-D scenes of vegetables placed on a table surface. The dataset has 6 style classes and 7 shape classes in total. iii) Replica dataset (Straub et al., 2019): it consists of 18 high quality reconstructions of indoor scenes. We use AI Habitat simulator (Manolis Savva* et al., 2019) to render multiview RGB-D data for it. We use the 152 instance-level shape categories provided by Replica. Due to lack of style labels, we manually annotate 16 style categories. Details on manual annotation process are present in the supplementary material. Figure 4 shows one example for both shape and style category.

We train D3DP-Nets self-supervised on posed multiview images in each dataset and learn the prototypes for each concept category. During training, we consider 1 and 5 labeled instances for each shape and style category in the dataset. During testing, we consider a pool of 1000 object instances.

In this experiment, we use ground-truth bounding boxes to isolate errors caused by different object detection modules. We compare D3DP-Nets with 2D, 2.5D and 3D versions of Prototypical Networks (Snell et al., 2017) that similarly classify object image crops by comparing object feature embeddings

Shape concept category: **Plant**    Style concept category: **Cream**

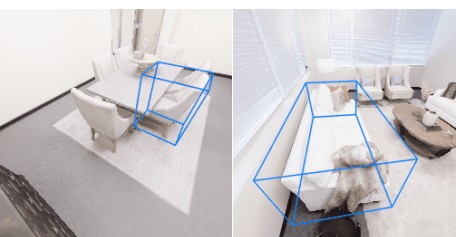

Figure 4: **Replica dataset.** On the left, we show two objects in different scenes belonging to the same shape cateogry 'Plant'. On the right, we show two objects belonging to the same style category 'Cream'.

| | CLEVR | | | | Real Veggie Data | | | | Replica | | | |
| | 5 shot | | 1 shot | | 5 shot | | 1 shot | | 5 shot | | 1 shot | |
| | Style | Shape | Style | Shape | Style | Shape | Style | Shape | Style | Shape | Style | Shape |
|---|---|---|---|---|---|---|---|---|---|---|---|---|
| D3DP-Net | **0.79** | 0.86 | **0.61** | 0.70 | **0.61** | 0.52 | **0.53** | 0.44 | **0.48** | 0.58 | **0.46** | **0.51** |
| 3DP-Net | 0.14 | 0.64 | 0.09 | 0.57 | 0.38 | 0.18 | 0.31 | 0.19 | 0.31 | 0.45 | 0.27 | 0.42 |
| 2D MUNIT | 0.50 | 0.54 | 0.41 | 0.47 | 0.43 | 0.48 | 0.39 | 0.38 | 0.30 | **0.60** | 0.23 | 0.42 |
| 2.5D MUNIT | 0.47 | 0.58 | 0.46 | 0.55 | 0.41 | 0.32 | 0.39 | 0.33 | 0.23 | 0.42 | 0.20 | 0.40 |
| GQN | 0.09 | 0.52 | 0.11 | 0.45 | 0.24 | 0.41 | 0.22 | 0.34 | 0.25 | 0.31 | 0.19 | 0.26 |
| D3D-Net | 0.43 | 0.48 | 0.26 | 0.40 | 0.31 | 0.28 | 0.18 | 0.24 | 0.23 | 0.29 | 0.10 | 0.14 |
| MB(Supervised) | 0.60 | **0.89** | 0.36 | **0.75** | 0.42 | **0.71** | 0.35 | **0.67** | 0.33 | 0.32 | 0.19 | 0.24 |

Table 1: Five & one shot classification accuracy for shape and style concepts in CLEVR (Johnson et al., 2017), Real Veggie, and Replica datasets.

to prototype embeddings. Specifically, we learn prototypical embeddings over the visual representations produced by the following baselines: (i) *2D MUNIT* (Huang et al., 2018) which disentangles shape and style within each object-centric 2D image RGB patch using the 2D equivalent of the shape-style disentanglement architecture of our model, and learns using an autoencoding objective (ii) *2.5D MUNIT* an extension of 2D MUNIT which uses concatenated RGB and depth as input. (iii) *3DP-Nets*, a version of D3DP-Nets where object shape-style disentanglement is omitted, this version corresponds to the scene representation learning model of Tung et al. (2019). (iv) Generative Query Network *GQN* of Eslami et al. (2016) which encodes multiview images of a scene and camera poses into a 2D feature map and is trained using cross-view prediction, similar to our model. (v) *D3D-Nets*, a version of D3DP-Nets where prototypical nearest neighbour retrieval is replaced with a linear layer which predicts the class probabilities. (iv) Meta-Baseline *MB* of Chen et al. (2020) is the SOTA *supervised* few-shot learning model, pre-trained using ImageNet. All baselines except *MB* are trained with the same unlabeled multiview image set as our method. All models classify each test image into a shape, and style category. Few-shot concept classification results are shown in Table 1. D3DP-Nets outperforms all unsupervised baselines. Interestingly, D3DP-Nets give better classification accuracy on the 1-shot task than almost all of the unsupervised baselines on the 5-shot task. Figure 5 shows a visualization of the style codes produced by D3DP-Nets (left) and 2.5D MUNIT baseline (right) on 2000 randomly sampled object instances from CLEVR using t-SNE (Maaten & Hinton, 2008). Each color represents a unique CLEVR style class. Indeed, in D3DP-Nets, codes of the same class are placed together, while for the *2.5D MUNIT* baseline, this is not the case.

## 4.2 FEW-SHOT VISUAL QUESTION ANSWERING

We integrate concept detectors built on the D3DP-Nets representation into modular neural networks for visual question answering, in which a question about an image is mapped to a computational graph over a small number of reusable neural modules including object category detectors, style detectors and spatial expression detectors. Specifically, we build upon the recent Neuro-Symbolic Concept Learner (NSCL) (Mao et al., 2019), as shown in Figure 3. In NSCL, the input and output of different neural modules are probability distributions over 2D object proposals denoting the probability that

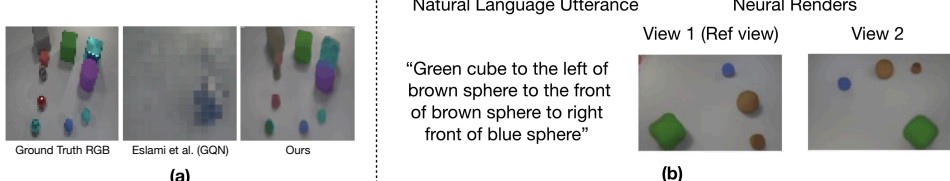

Figure 6: (a) View predictions of D3DP-Nets and GQN (Eslami et al., 2018) in novel scenes. (b) Generating novel scenes using only a single example for each style and shape category.

the executed subprogram is referring to each object, and their object category, color and material classifiers also use nearest neighbors over learnt prototypes. For example, in the question *"How many yellow objects are there?"*, the model first uses the color classifier to predict for all objects the probability that they are yellow, and then uses the resulting probability map to give an answer. NSCL learns 1D prototypes for object shape, color and material categories and classifies objects to labels using nearest neighbors to these prototypes. In our D3DP-Nets-VQA architecture, we have 3D instead of 2D object proposals, and disentangled 3D shape and 1D color/material and spatial relationship prototypes instead.

We compare D3DP-VQA against the following models: i) *NSCL-2D* (with and without ImageNet pretraining), the state of the art model of Mao et al. (2019) that uses a ResNet-34 pretrained on ImageNet as input feature representations ii) *NSCL-2.5D*, in which the object visual representations for shape/color/material are computed over RGB and depth concatenated object patches as opposed to RGB alone. This model is pretrained with a view prediction loss using the CLEVR dataset in Sec. 4.1 iii) *NSCL-2.5D-disentangle* that uses disentangled object representations generated by our 2.5D MUNIT disentangling model, iv) *D3DP without 3D shape prototypes*, a version of D3DP-

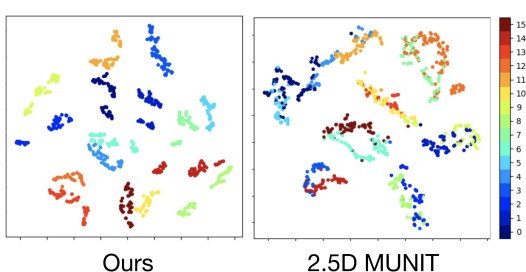

Figure 5: t-SNE visualization on styles codes.

Nets that replaces the 3-dimensional shape codes with 1D ones obtained by spatial pooling v) *D3DP without disentanglement*, that learns prototypes for shape, color and material on top of entangled 3D tensors.

We consider the same supervision for our model and baselines in the form of densely annotated scenes with object attributes and 3D object boxes. We use ground-truth neural programs so as to not confound the results with the performance of a learned parser. More details on the VQA experimental setup and additional ablative experiments are included in the supplementary file.

VQA performance results are shown in Table 2. We evaluate by varying the number of training scenes from 10 to 250. For each training scene we generate 10 questions. The original CLEVR dataset included 70,000 scenes and 700,000 questions, so even when training with 250 scenes, we are training with 0.35% the number of original scenes. Our full model outperforms all of the alternatives, showing the importance of both the 3D feature representations as well as disentanglement of shape and style. To test our model's one shot generalization ability on questions about object categories it had not seen in the original training set, we introduce a new test set consisting of only novel objects. We generate a test set of 500 scenes in the CLEVR environment with three new objects: "cheese", "garlic", and "pepper" and introduce them to our model and baselines using one example image of each, associated with its shape category label. We provide example scene/question pairs for this setting in the supplementary. The results described in Table 2 indicate that our model is able to maintain its ability to answer questions even when seeing completely novel objects and with very few training examples. The SOTA 2D model outperforms our model on the in domain test set because it is able to exploit pretraining on ImageNet, which our models are unable to do. However, our model is able to adapt much better than both the 2D and 2.5D baselines in few-shot regime.

| VQA Model | In domain test set | | | | | One shot test set | | | | |
|---|---|---|---|---|---|---|---|---|---|---|
| | Number of Training Examples | | | | | Number of Training Examples | | | | |
| | 10 | 25 | 50 | 100 | 250 | 10 | 25 | 50 | 100 | 250 |
| Our full model | **0.809** | 0.872 | 0.902 | 0.923 | 0.939 | **0.775** | **0.836** | **0.834** | 0.828 | 0.845 |
| without 3D shape prototypes | 0.798 | 0.858 | 0.538 | 0.905 | 0.932 | 0.410 | 0.410 | 0.517 | 0.745 | 0.771 |
| without shape/style disentanglement | 0.458 | 0.407 | 0.616 | 0.806 | 0.788 | 0.457 | 0.402 | 0.616 | 0.807 | 0.792 |
| without 3D shape prototypes and without shape/style disentanglement | 0.718 | 0.829 | 0.849 | 0.868 | 0.894 | 0.608 | 0.681 | 0.688 | 0.692 | 0.701 |
| Entangled disentangled features | 0.648 | 0.565 | 0.899 | 0.917 | 0.928 | 0.619 | 0.542 | 0.812 | 0.831 | 0.813 |
| InstanceNorm disentangled features + rotation-aware check | 0.606 | 0.831 | 0.875 | 0.894 | 0.905 | 0.627 | 0.775 | 0.832 | **0.836** | **0.861** |
| 2D NSCL Mao et al. (2019) | 0.733 | **0.927** | **0.959** | **0.978** | **0.990** | 0.594 | 0.708 | 0.703 | 0.789 | 0.743 |
| 2D NSCL Mao et al. (2019) without ImageNet pretraining | 0.514 | 0.624 | 0.682 | 0.844 | 0.931 | 0.467 | 0.502 | 0.553 | 0.624 | 0.679 |
| 2.5D NSCL Mao et al. (2019) | 0.594 | 0.737 | 0.828 | 0.881 | 0.925 | 0.528 | 0.633 | 0.651 | 0.633 | 0.633 |
| 2.5D NSCL Mao et al. (2019) disentangled | 0.436 | 0.486 | 0.640 | 0.735 | 0.842 | 0.430 | 0.462 | 0.517 | 0.561 | 0.564 |

Table 2: VQA results with model compared to ablations and baselines.

### 4.3 View prediction

We qualitatively compare our model with the Generative Query Network (GQN) of Eslami et al. (2016) on the task of view prediction in Figure 6 (a). The figure shows view prediction results for a scene with more objects than those at training time. D3DP-Nets dramatically outperforms GQN, which we attribute to the 3-dimensional representation bottleneck that better represents the 3D space of the scene, as compared to the 2D bottleneck of Eslami et al. (2016).

### 4.4 3D scene generation from language utterances

We test D3DP-Nets on the task of scene generation from language utterances. Given annotations for each prototype, our model can generate 3D scenes that comply with a language utterance, as seen in Figure 12 (b). We assume the parse tree of the utterance is given. Our model generates each object's 3D feature map by combining shape and style prototypes as suggested by the utterance, and placing them iteratively on a background canvas making sure it complies with the spatial constraints in the utterance (Prabhudesai et al., 2019). Our model can generate shape and style combinations not seen at training time. We neurally render the feature map to generate the RGB image.

## 5 Conclusion and Future Directions

We presented D3DP-Nets, a model that learns disentangled 3D representations of scenes and objects and distills them into 3D and 1D prototypes of shapes and styles using multiview RGB-D videos of static scenes. We trained classifiers of prototypical object categories, object styles, and spatial relationships, on disentangled relevant features. We showed that they generalize better than 2D representations or 2D disentangled representations, with less training data. We showed modular architectures for VQA over our concept classifiers permit few-shot generalization to scenes with novel objects. We hope our work will stimulate interest in self-supervising 3D feature representation for 3D visual recognition and question answering in domains with few human labels. Finally adding deformation to our model which will permit a prototype to match against more instances of the same class and expanding to diverse natural language datasets of VQA which will require us to add more diverse programs are direct avenues of future work.

## 6 Acknowledgements

This work has been partly funded by Sony AI, DARPA Machine Common Sense and the Air Force Office of Scientific Research under award number FA9550-20-1-0423. Any opinions, findings, and conclusions or recommendations expressed in this material are those of the author(s) and do not necessarily reflect the views of the United States Air Force.

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

# A    DATASET PREPARATION

**CLEVR Dataset.**    We construct four different datasets using the CLEVR Blender simulator as a base (Johnson et al., 2017). One key difference from the original simulator is that we generate RGB-D images of scenes instead of RGB images.

The first dataset is a support dataset containing 1200 scenes in the training split and 400 scenes in the validation split. For each scene, 12 different RGB-D views are generated (4 different azimuths, 3 different elevations). The azimuths are in 90° increments and the elevations are 20°, 40°, and 60°. This dataset is used in the unsupervised training of the D3DP-Nets.

The second dataset generated contains 5000 scenes in the training split and 1500 scenes in each of the validation and testing splits, with 10 questions generated for each scene. Each scene is rendered according to the specification of the original CLEVR dataset. This dataset is used for the training of all VQA models. Examples can be seen in Figure 7.

The third and fourth datasets are used to test the one-shot learning capabilities of the VQA models. They introduce three new shapes: *cheese*, *garlic*, *pepper*. The prototype split for both datasets contains one scene for each object with a single viewpoint rendered for each scene. The scene for each object contains that object centered on floor with the large size and random color and material. The test split for the third dataset contains 500 scenes with a mix of the shapes seen during training and novel shapes. The test split for the fourth dataset contains 500 scenes with only the novel shapes. Each scene in the test split is rendered from a single viewpoint (approximately the same as the viewpoint used in the second dataset) and has 10 questions generated about it. Examples from the novel only dataset can be seen in Figure 8. The results for the novel only dataset are shown in the main paper, and the results for the mixed dataset are shown in Appendix E.

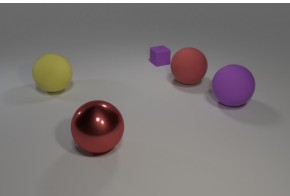 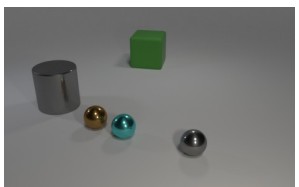 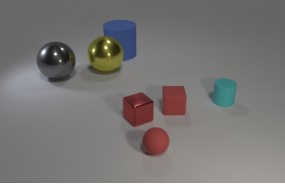

**Q:** There is a purple matte object in front of the red object behind the rubber thing in front of the yellow matte ball; what shape is it?
**A:** sphere

**Q:** What number of tiny shiny balls have the same color as the cylinder?
**A:** 1

**Q:** Does the tiny cyan thing have the same material as the large blue cylinder?
**A:** True

Figure 7: Example scene/QA pairs from the training dataset used for VQA

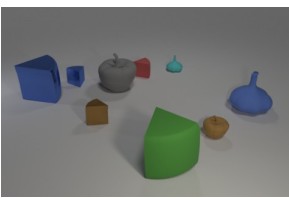 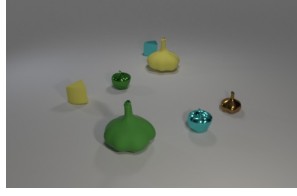 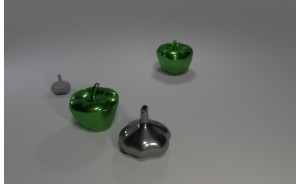

**Q:** Do the cyan matte thing and the matte cheese that is in front of the brown metallic cheese have the same size?
**A:** False

**Q:** How many cheeses are either small gray shiny objects or small objects?
**A:** 2

**Q:** What material is the other gray thing that is the same shape as the tiny matte object?
**A:** metal

Figure 8: Example scene/QA pairs from the novel only one shot test dataset used for VQA.

**CARLA Dataset.** We use CARLA dataset to show detector improvement results in Appendix D. We use the 26 vehicle classes available in Carla 0.9.7 to prepare our dataset. We call each recorded datapoint a scene. Each scene consists of multiview RGB and depth images of static vehicles placed at randomly selected spawn locations. We generate two separate datasets: datasets with scenes consisting of only single vehicles and datasets with scenes consisting of 3-6 vehicles. The single vehicle scenes are used to train the 3D detector. The multi-vehicle scenes are used to evaluate the trained 3D detector. For clarity, we will use $a = Uniform(b, c)$ to mean that 'a' takes a random value between 'b' and 'c'. For each scene, we first randomly select a map from the available CARLA maps. Then we randomly perturb the weather conditions by setting $cloudiness = Uniform(0, 70)$, $precipitation = Uniform(0, 75)$, and $sun\_altitude\_angle = Uniform(30, 90)$. We place a total of 18 RGB-D cameras in the scene. The origin of a selected *player vehicle* serves as the origin with respect to which the extrinsics of all the cameras is calculated. For single-vehicle scenes, we randomly select a spawn point and place the vehicle there. This vehicle then also acts as the *player vehicle*. For multi-vehicle scenes, we again randomly select a spawn point, place a vehicle there, and mark it as the *player vehicle*. Then, we randomly determine how many more vehicles to place. We then find all spawn points that are within 17 units distance from the first spawn point and randomly select some of them to place the extra vehicles. Selecting nearby spawn locations helps ensure that most of the vehicles will be visible in the majority of the camera views. For vehicles in CARLA, the x-axis points forward, the y-axis points to the right, and the z-axis points upwards. For all scenes, we place the first camera at $x = Uniform(8, 13)$, $y = 0$, $z = 1$, $pitch = 0°$, $yaw = -180°$, $roll = 0°$. The next 8 cameras are placed on the boundary of a circle centered at the *player vehicle*, with $radius = Uniform(7, 14)$ (radius is randomly sampled for each camera position), $z = 5.5$, $pitch = 0°$, $roll = 0°$, and $yaw$ decremented uniformly by $35°$ from $-40°$ to $-285°$. The next 8 cameras follow the same setup but with $pitch = -40°$ and $z = 6.5$. Finally, the last RGB-D camera is placed overhead at $x = 0$, $y = 0$, $z = Uniform(6, 10)$, $pitch = -90°$, $yaw = 0°$, and $roll = 0°$.

**Real World Veggie Dataset.** Our real world veggie dataset consists of multiview RGB-D scenes of vegetables placed on a table recorded with Microsoft Azure Kinect camera. For each scene, we place 1 to 6 vegetables on a table and move the camera around the table randomly to capture RGB and depth images. A SLAM package then takes the camera output and provides us with the extrinsics. To get the 3D bounding boxes of objects in the scene, we fire a 2D object detector on each RGB image and triangulate the 2D detections to generate the 3D bounding boxes.

**Replica Dataset.** Replica dataset (Straub et al., 2019) provides high quality reconstructions for 18 indoor scenes. We use AI Habitat simulator (Manolis Savva* et al., 2019) to render multiview RGB-D data for these meshes. Specifically, to capture one scene, we load a random mesh in the simulator, select one object (primary object) in the mesh randomly, and spawn the agent near that object. We then move the agent around that object, ensuring that agent is between 1m to 2m from the primary object, and capture RGB-D images from 24 different viewpoints. We use $256 \times 256$ spatial resolution for RGB and depth images. These images can also include objects in the vicinity of the primary object we selected. For a particular view, we only store information for objects that occupy more than 500 pixels in the RGB image for that view. We also manually annotate the style and shape category for each object visible in any of the 24 views. For shape category, we use the instance id provided by the AI Habitat simulator. This gives us 152 shape categories. For style, we annotate each object with a color and use the color label as style for that object. This gives us 16 different style categories.

# B   ARCHITECTURE DETAILS

For all the models, $E^{sc}$ and $D^{sc}$ trained over view prediction losses are used as the base model.

The input RGB and depth images are resized to a resolution of $320 \times 480$ for all the datasets. While training using view prediction, we randomly sample 2 views from each multi-view scene. During testing, we only use a single view. Our model converges in 10-12hrs of training and requires 0.8 seconds for an inference step on a single RTX 2080.

### B.1 D3DP-NETS

**2.5D-to-3D Lifting** ($\mathrm{E^{sc}}$) **and 3D-to-2.5D Projection** ($\mathrm{D^{sc}}$ **and** $\mathrm{D^{occ}}$**).** Our 2.5D-to-3D lifting, 3D occupancy estimation and 2D RGB estimation modules follow the exact same architecture as Harley et al. (2020). However, unlike their architecture, we do not use batch normalization in our network because we did not find it to be compatible with adaptive instance normalization which is used later in the pipeline. Our 2D-to-3D Lifting module takes as input RGB-D images, camera intrinsics, and camera extrinsics and outputs a 3D feature map of size $72 \times 72 \times 72 \times 32$, where 32 is the number of channels and 72 is the height, width, and depth. We explain the implementation details of each of these modules below.

- **2.5D-to-3D lifting** Our 2.5D-to-3D unprojection module takes as input RGB-D images and converts it into a 4D tensor $\mathbf{U} \in \mathbb{R}^{w \times h \times d \times 4}$, where $w, h, d$ is 72, 72, 72. We use perspective (un)projection to fill the 3D grid with samples from 2D image. Specifically, using pinhole camera model (Hartley & Zisserman, 2003), we find the floating-point 2D pixel location that every cell in the 3D grid, indexed by the coordinate $(i, j, k)$, projects onto from the current camera viewpoint. This is given by $[u, v]^T = \mathbf{KS}[i, j, k]^T$, where $\mathbf{S}$, the similarity transform, converts memory coordinates to camera coordinates and $\mathbf{K}$, the camera intrinsics, convert camera coordinates to pixel coordinates. Bilinear interpolation is applied on pixel values to fill the grid cells. We obtain a binary occupancy grid $\mathbf{O} \in \mathbb{R}^{w \times h \times d \times 1}$ from the depth image $\mathbf{D}$ in a similar way. This occupancy is then concatenated with the unprojected RGB to get a tensor $[\mathbf{U}, \mathbf{O}] \in \mathbb{R}^{w \times h \times d \times 4}$. This tensor is then passed through a 3D encoder-decoder network, the architecture of which is as follows: 4-2-64, 4-2-128, 4-2-256, 4-0.5-128, 4-0.5-64, 1-1-$F$. Here, we use the notation $k$-$s$-$c$ for kernel-stride-channels, and $F$ is the feature dimension, which we set to $F = 32$. We concatenate the output of transposed convolutions in decoder with same resolution feature map output from the encoder. The concatenated tensor is then passed to the next layer in the decoder. We use leaky ReLU activation after every convolution layer, except for the last one in each network. We obtain our 3D feature map $\mathbf{M}$ as the output of this process.

- **3D occupancy estimation.** In this step, we want to estimate whether a voxel in the 3D grid is "occupied" or "free". The input depth image gives us partial labels for this. We voxelize the pointcloud to get sparse "occupied" labels. All voxel cells that are intersected by the ray from the source-camera to each occupied voxel are marked as "free". We give $\mathbf{M}$ as input to the occupancy module. It produces a new tensor $\mathbf{C}$, where each voxel stores the probability of being occupied. We use a 3D convolution layer with a $1 \times 1 \times 1$ filter followed by a sigmoid non-linearity to achieve this. We train this network with the logistic loss, $\mathcal{L}_{\mathrm{occ}} = (1/\sum \hat{\mathbf{I}}) \sum \hat{\mathbf{I}} \log(1 + \exp(-\hat{\mathbf{C}} \cdot \mathbf{C}))$, where $\hat{\mathbf{C}}$ is the label map, and $\hat{\mathbf{I}}$ is an indicator tensor, indicating which labels are valid. Since there are far more "free" voxels than "occupied", we balance this loss across classes within each minibatch.

- **2D RGB estimation.** Given a camera viewpoint $v_q$, this module projects the 3D feature map $\mathbf{M}$ to "render" 2D feature maps. To achieve this, we first obtain a view-aligned version, $\mathbf{M}_{v_q}$, by resampling $\mathbf{M}$. The view oriented tensor, $\mathbf{M}_{v_q}$, is then warped so that perspective viewing rays become axis-aligned. This gives us the perspective-transformed tensor $\mathbf{M}_{proj_q}$. This tensor is then passed through a CNN to get a 2D feature map $v_q$. The CNN has the following architecture (using the notation $k$-$s$-$c$ for kernel-stride-channels): max-pool along the depth axis with $1 \times 8 \times 1$ kernel and $1 \times 8 \times 1$ stride, to coarsely aggregate along each camera ray, 3D convolution with 3-1-32, reshape to place rays together with the channel axis, 2D convolution with 3-1-32, and finally 2D convolution with 1-1-$E$, where $E$ is the channel dimension, $E = 3$.

**Object-Centric Encoder** $\mathrm{E^o}$ Our object encoder $\mathrm{E^o}$ consists of two sub encoders, the content encoder $\mathrm{E}_{shp}$ and the style encoder $\mathrm{E}_{sty}$. Both $\mathrm{E}_{shp}$ and $\mathrm{E}_{sty}$ take as input an object centric 3D feature map of size $16 \times 16 \times 16 \times 32$. $\mathrm{E}_{sty}$ uses two 3D convolutions with the following architecture (our notation is k-s-c-p-pt for kernel size, stride, output channels, padding, and padding type): 3-1-64-1-constant, 4-2-128-1-constant. Each 3D convolution operation is followed by ReLU non-linearity. The output of the second ReLU is averaged pooled spatially to get a linear output, which is the style code. This style code is then passed through two linear layers, both producing 256 dimensional output.

We use ReLU non-linearity only after the first linear layer. The final output of the linear layer is used as adaptive instance normalization (AdaIN) parameters to re-normalize the intermediate features generated by D. $E_{shp}$ also uses two 3D convolution layers with the following architecture: 3-1-64-1-constant, 4-2-128-1-constant. Both convolution operations are followed by Instance Normalization (Ulyanov et al., 2016) and ReLU non-linearity.

**Object-Centric Decoder** D    The object-centric decoder D takes as input the shape and style codes from $E_{shp}$ and $E_{sty}$, and compose them back into a complete object-centric feature maps. It consists of two 3D convolution layers with the architectures: 3-1-128-1-constant, 3-1-32-1-constant. The output of the first 3D convolution is re-normalized using the AdaIN parameters generated by $E_{sty}$, before passing through a ReLU non-linearity. This output is then upsampled by a factor of 2 and passed through the second 3D convolution. The output of D is an object centric feature map with content and style dictated by the inputs of $E_{shp}$ and $E_{sty}$ respectively.

**3D Detector**    We use the same 3D detector architecture as in Tung et al. (2019). This 3D detector extends the Faster RCNN architecture (Ren et al., 2015) to use 3D feature maps to predict 3D bounding boxes, instead of predicting 2D bounding boxes using 2D feature maps. The output of our 2D-to-3D Lifting module, with size $72 \times 72 \times 72 \times 32$, is fed as input to the 3D detector. The 3D detector consists of one down-sampling layer and three 3D residual blocks. Each layer has 32 channels. At each grid location in the 3D feature map, we use a single cube shaped anchor box of side length 12 units.

## B.2    VISUAL QUESTION ANSWERING (VQA)

The VQA model architecture is similar to the Neuro-Symbolic Concept Learner (NSCL) (Mao et al., 2019). We assume each object in a scene can have certain attributes and each pair of objects can have a relationship associated with it. For our tasks, the object attributes are *shape*, *color*, *material*, and *size*; the relationships are all *spatial*. The model has a neural operator for each of the possible attributes/relationships. These operators take as input features corresponding to a specific instance of an object/relationship. The outputs of the operators are compared to a dictionary of embeddings representing specific concepts (e.g. *red* or *sphere*) to compute the probability that the instance matches the concept. We do this by computing the cosine similarity of the computed embeddings with the stored embeddings, and using those similarities as logits in our probability calculation. For the shape attribute, our embeddings are 4D, while for the others, the embeddings are 1D. This allows us to do a rotation aware similarity calculation for shape, as described in the main paper.

For every object in the scene, five disentangled feature sets are created: shape and style from the D3DP-Nets and center location, size, and rotation from the bounding box description. For every pair of objects, the relationship feature sets are created by concatenating the corresponding feature sets of both objects (e.g. content with content).

Since the feature sets we input into the operators are disentangled, we can feed specific inputs into each operator. The feature set descriptions and the usage of the feature sets by each attribute operator are shown in Table 3. The 1D feature sets are first passed through a linear layer with output dimension 256. The content feature set is passed through a 3-1-256-1-constant 3D convolution. The relevant feature set is then passed to each attribute operator. For the shape attribute, we use two 3D convolution layers to produce the final embedding: 1-1-256-0-constant and 1-1-64-0-constant. For the other attributes, we use a linear layer with output dimension 256 and a linear layer with output dimension 64. Every layer described above except for the final attribute operator layers is followed by a ReLU non-linearity. These embeddings are the ones compared to the stored concept embeddings in the model's dictionary. Every VQA model is trained for 60 epochs with early stopping. We use the Adam optimizer (Kingma & Ba, 2014) initialized with a learning rate of .001.

We also trained a semantic parser to be able to answer questions for which we do not have a symbolic program generated. The architecture of this parser follows the architecture described in Mao et al. (2019). Instead of training it with reinforcement learning during the VQA training, we train the parser separately in a supervised manner with teacher forcing using a small set of question, program pairs. The accuracy of this parser is 94%, allowing us to use it for most questions that are of similar form to those in the CLEVR dataset.

| Feature set | Shape | Description |
|---|---|---|
| D3DP-Nets Content | 8x8x8x128 | The content output of D3DP-Nets |
| D3DP-Nets Style | 128 | The style output of D3DP-Nets |
| Bounding box center | 3 | The $(x, y, z)$ location of the center of the bounding box containing the object |
| Bounding box size | 3 | The length, width, and height of the bounding box containing the object |
| Bounding box rotation | 3 | The pitch, roll, and yaw rotations of the bounding box containing the object |

| Operator | Input (in final model) | 3D Embedding Used |
|---|---|---|
| Shape | D3DP-Nets Content | Yes |
| Color | D3DP-Nets Style | No |
| Material | D3DP-Nets Style | No |
| Size | Bounding box-size | No |
| Spatial relationship | Bounding box-centers | No |

Table 3: Left: Description of feature sets available during VQA. Right: Input feature sets used by each module.

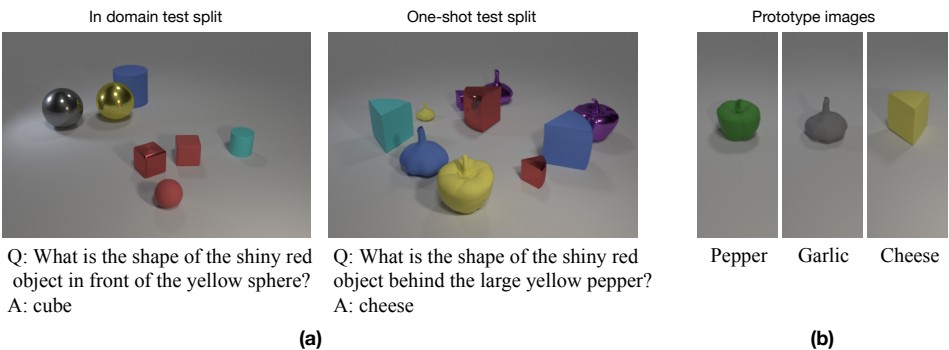

Q: What is the shape of the shiny red object in front of the yellow sphere?
A: cube

Q: What is the shape of the shiny red object behind the large yellow pepper?
A: cheese

(a)

Pepper    Garlic    Cheese

(b)

Figure 9: **(a)** The left scene/question pair is from the in domain test set, and the right scene/question pair is from the one shot test set. The colors, materials, sizes, and spatial relationships tested in both splits are the same. The only difference is that the one shot test set contains shapes the model did not see while training and was only exposed to one example before the testing phase. **(b)** The prototype images shown to the model before starting the one shot testing phase.

In figure 9, we show example scene/question pairs for the in domain test set and one shot test set.

## C   FEATURE SUBSPACE SELECTION USING CONTRASTIVE EXAMPLES

In this section we show that, given contrastive examples, our disentangled representation can be used to infer relevant feature subspace for each concept category. We do this by selecting the feature subspace which has the minimum cosine similarity given a contrastive example. For example, in order to find what input we should give to Color module, we take 3D object centric feature maps for two objects differing only in their color. We then disentangle the feature maps into content and style features. We finally calculate the cosine similarity between corresponding content and style features for both the objects and use the features which has lower similarity. If our disentanglement is good, we should be able to infer correct feature subspace for different concept categories, as changes in one feature subspace should leave the other disentangled feature subspaces unchanged. In Table 5 we show the accuracy of selecting the correct feature subspace given 10 contrastive examples for each concept category. For retrieving a cosine similarity friendly feature subspace for size and spatial relation categories, we train an auto-encoder on top of their inputs, and use their encoded representation as the relevant feature subspace. In Figure 11, we show contrastive examples for each concept category.

## D   LEARNING 3D OBJECT DETECTORS BY GENERATION

Upon training, D3DP-Nets  maps a novel RGB-D image to a set of 3D object boxes, their shape and style codes, and the background scene feature map. We generate (simulate) novel 3D scene feature maps by adding object feature maps against background scene maps while randomizing object 3D locations, 3D pose and 3D sizes. We make sure each added 3D object box does not intersect in 3D with the predicted scene occupancy $M^{occ}$ and objects added thus far. We train our 3D object detectors using additional annotations from such labelled imagined scene feature maps. We show below that such mental augmentations are beneficial for generalization of 3D object detector from

| Annotation Source | CARLADosovitskiy et al. (2017) | | | | | | CLEVRJohnson et al. (2017) | | | | | |
|---|---|---|---|---|---|---|---|---|---|---|---|---|
| | 50 annot. | | 100 annot. | | 200 annot. | | 50 annot. | | 100 annot. | | 200 annot. | |
| | IOU .3 | IOU .5 | IOU .3 | IOU .5 | IOU .3 | IOU .5 | IOU .3 | IOU .5 | IOU .3 | IOU .5 | IOU .3 | IOU .5 |
| Real | 0.57 | 0.41 | 0.57 | 0.44 | 0.60 | 0.47 | 0.43 | 0.43 | 0.47 | 0.44 | 0.50 | 0.48 |
| Real + Aug | **0.61** | **0.45** | **0.67** | **0.55** | **0.64** | **0.50** | **0.44** | **0.46** | **0.50** | **0.47** | **0.54** | **0.50** |

Table 4: Mean average precision for category agnostic region proposals.

| Shape | Color | Material | Spatial Relation | Size |
|---|---|---|---|---|
| 0.9 | 1.0 | 1.0 | 1.0 | 1.0 |

Table 5: Feature selection accuracy on CLEVR dataset given 10 randomly selected contrastive examples per category.

few examples, and can generate improvements, especially in the low IoU evaluation regime. Note that D3DP-Nets's imaginations do not attempt to match the training scene distribution. Rather, they target combinatorial generalization (Battaglia et al., 2018), using basic spatial reasoning of free space encoded in the 3D neural representations. In Figure 10, we visualize RGB neural renders of our generated novel 3D feature maps. We generate (simulate) novel 3D scene feature maps $\mathbf{M}^{sim}$ by adding entangled object feature maps against background scene maps while randomizing object 3D locations and 3D sizes as shown in Figure 1. We consider 50, 100 and 200 annotations of 3D bounding box on single object scenes in CARLA and CLEVR datasets. We use those annotations to train corresponding category agnostic 3D object detectors. D3DP-Nets weights are initialized with self-supervised view-prediction using a support dataset made up of 1600 scenes with 12 views each (4 different azimuths, 3 different elevations). Implementation details of our 3D detector architecture can be found in the supplementary file. We generate neural scene imaginations following the method described above, and use it as additional training data. Specifically, we consider randomly placing 3-10 objects around the (real) object in the inferred 3D scene maps of the training images. We test the model with 300 scenes from each simulated environment, with each having around 3-8 objects present. We compare a 3D detector trained on joint real and hallucinated data, with one trained on real data alone. We use L2 weight regularization in all methods, and cross-validate the weight decay. We use an early stopping technique to avoid overtraining of the detector. In Table 4, we show that training the 3D detector on real and hallucinated neural scenes outperforms just using real images alone. We found that most of our improvement in mean AP comes from placing more objects and randomizing their location and not from composing unique shape-style. We believe this is because our entangled object representation generated from unique shape/style composition is distinguishable from the actual real object, which the detector exploits.

## E  VQA Additional Results

**Further ablations**    We introduce three more ablations of our VQA model that were not presented above due to space considerations.

The first ablation entangles all the disentangled feature sets together before feeding them into the attribute operators. It does so by concatenating the different feature sets after upsampling/downsampling to match dimensions, and then passing through either a linear layer or 3D convolution layer depending on if the input/desired output embedding is 1D or 4D.

The second ablation involves using a simple method to extract content and style embeddings. We take the input 3D tensors, normalize by channel to get the content embedding, concatenate the channel means and channel standard deviations to get the style embedding. This simple baseline performs surprisingly well at the higher data settings we test. The full results for both ablations are shown in Table 6.

The final ablation removes all disentanglement and 3D prototypes. This model is pretrained only with view prediction, and does not do any shape/style disentanglement. It also pools any 3D prototypes to 1D prototypes before comparison instead of doing a rotation aware check.

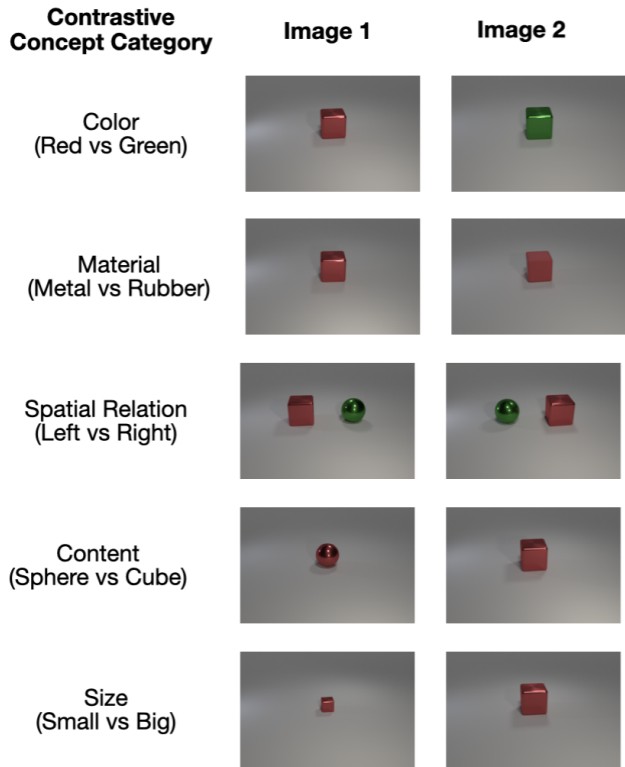

Figure 11: Contrasting examples for each concept category. First column specifies the concept and the contrasting attributes shown for that concept. Next two columns show the images differing only in that specified concept.

**Attribute classifier accuracy** We present the classification accuracy of the attribute modules for the VQA experiments in this paper in Tables 7, 8, 9, and 10.

For the shape classifier in particular, we find that our full model is worse than the 2D baseline when tested on in domain examples, but when tested on the one shot dataset, the shape classification accuracy of the 2D baseline decreases sharply. The 3D models meanwhile are able to maintain decent performance. Our full model does not have the best shape classification accuracy on the one shot dataset, but it still shows a respectable performance. Finally, we note our model tended to do worse on the shape classification accuracy when trained with more examples in the training phase. This phenomenon requires further exploration, but a possible explanation could be that more training examples of the original shapes encodes stronger biases into the model, making it more difficult to identify the one shot objects.

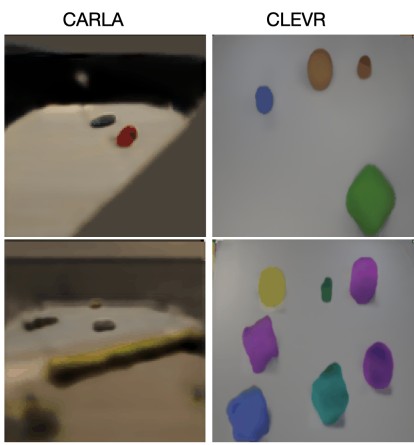

Figure 10: RGB neural renders of the novel 3D scene feature maps generated by our imagination module. Although we don't get pixel accurate generation, our synthesized 3D feature map encodes the semantic structure of the scene.

**Mixed one shot only dataset** We present results on the mixed one shot test dataset introduced in Appendix A. This dataset contains a mix of objects seen in training and novel objects. Table 11 shows results on this dataset.

| VQA Model | In domain test set | | | | | One shot test set | | | | |
|---|---|---|---|---|---|---|---|---|---|---|
| | Number of Training Examples | | | | | Number of Training Examples | | | | |
| | 10 | 25 | 50 | 100 | 250 | 10 | 25 | 50 | 100 | 250 |
| D3DP | **0.809** | 0.872 | 0.902 | 0.923 | 0.939 | **0.775** | **0.836** | **0.834** | **0.828** | **0.845** |
| D3DP without 3D shape prototypes | 0.798 | 0.858 | 0.538 | 0.905 | 0.932 | 0.410 | 0.410 | 0.517 | 0.745 | 0.771 |
| D3DP without shape/style disentanglement | 0.458 | 0.407 | 0.616 | 0.806 | 0.788 | 0.457 | 0.402 | 0.616 | 0.807 | 0.792 |
| NSCL-2D (Mao et al., 2019) | 0.733 | **0.927** | **0.959** | **0.978** | **0.990** | 0.594 | 0.708 | 0.703 | 0.789 | 0.743 |
| NSCL-2.5D (Mao et al., 2019) | 0.594 | 0.737 | 0.828 | 0.881 | 0.925 | 0.528 | 0.633 | 0.651 | 0.633 | 0.633 |
| NSCL-2.5D-disentangled (Mao et al., 2019) | 0.436 | 0.486 | 0.640 | 0.735 | 0.842 | 0.430 | 0.462 | 0.517 | 0.561 | 0.564 |

Table 6: VQA performance of our model and baselines in CLEVR (Johnson et al., 2017) under a varying number of annotated training scenes.

| VQA Model | In domain test set | | | | | One shot test set | | | | |
|---|---|---|---|---|---|---|---|---|---|---|
| | Number of Training Examples | | | | | Number of Training Examples | | | | |
| | 10 | 25 | 50 | 100 | 250 | 10 | 25 | 50 | 100 | 250 |
| Our full model | **0.772** | 0.828 | 0.856 | 0.875 | 0.904 | 0.808 | 0.814 | 0.766 | 0.729 | 0.740 |
| without 3D shape prototypes | 0.762 | 0.793 | 0.734 | 0.857 | 0.899 | 0.529 | 0.649 | 0.535 | 0.557 | 0.600 |
| without shape/style disentanglement | 0.350 | 0.326 | 0.441 | 0.698 | 0.692 | 0.229 | 0.426 | 0.476 | **0.870** | 0.879 |
| without 3D shape prototypes and without shape/style disentanglement | 0.682 | 0.730 | 0.775 | 0.795 | 0.818 | 0.422 | 0.409 | 0.416 | 0.418 | 0.421 |
| Entangled disentangled features | 0.492 | 0.724 | 0.845 | 0.859 | 0.893 | 0.459 | 0.639 | 0.766 | 0.855 | 0.911 |
| InstanceNorm disentangled features + rotation-aware check | 0.733 | 0.803 | 0.834 | 0.841 | 0.835 | **0.818** | **0.870** | **0.850** | 0.836 | **0.894** |
| 2D NSCL Mao et al. (2019) | 0.725 | **0.958** | **0.980** | **0.991** | **0.996** | 0.481 | 0.574 | 0.596 | 0.664 | 0.632 |
| 2D NSCL Mao et al. (2019) without ImageNet pretraining | 0.479 | 0.661 | 0.727 | 0.804 | 0.899 | 0.263 | 0.131 | 0.395 | 0.381 | 0.426 |
| 2.5D NSCL Mao et al. (2019) | 0.527 | 0.662 | 0.756 | 0.809 | 0.881 | 0.206 | 0.410 | 0.455 | 0.335 | 0.342 |
| 2.5D NSCL Mao et al. (2019) disentangled | 0.707 | 0.768 | 0.838 | 0.884 | 0.930 | 0.464 | 0.534 | 0.496 | 0.438 | 0.406 |

Table 7: Accuracy of the shape classifiers of the VQA models

| VQA Model | In domain test set | | | | | One shot test set | | | | |
|---|---|---|---|---|---|---|---|---|---|---|
| | Number of Training Examples | | | | | Number of Training Examples | | | | |
| | 10 | 25 | 50 | 100 | 250 | 10 | 25 | 50 | 100 | 250 |
| Our full model | **0.949** | **0.970** | 0.978 | 0.981 | 0.983 | **0.899** | 0.952 | 0.965 | 0.976 | 0.972 |
| without 3D shape prototypes | 0.948 | **0.970** | 0.828 | 0.981 | 0.982 | 0.112 | 0.125 | 0.225 | 0.974 | 0.976 |
| without shape/style disentanglement | 0.158 | 0.129 | 0.952 | 0.982 | 0.983 | 0.149 | 0.122 | 0.926 | 0.967 | 0.960 |
| without 3D shape prototypes and without shape/style disentanglement | 0.833 | 0.969 | 0.979 | 0.982 | 0.981 | 0.733 | 0.920 | 0.963 | 0.971 | 0.968 |
| Entangled disentangled features | 0.836 | 0.483 | 0.979 | 0.982 | 0.984 | 0.759 | 0.401 | 0.973 | 0.974 | 0.975 |
| InstanceNorm disentangled features + rotation-aware check | 0.645 | 0.962 | 0.974 | 0.979 | 0.982 | 0.578 | 0.895 | 0.964 | 0.962 | 0.960 |
| 2D NSCL Mao et al. (2019) | 0.827 | 0.945 | **0.980** | **0.989** | **0.992** | 0.820 | 0.936 | **0.988** | **0.996** | **0.993** |
| 2D NSCL Mao et al. (2019) without ImageNet pretraining | 0.229 | 0.451 | 0.529 | 0.930 | 0.980 | 0.204 | 0.353 | 0.444 | 0.872 | 0.976 |
| 2.5D NSCL Mao et al. (2019) | 0.669 | 0.880 | 0.969 | 0.977 | 0.984 | 0.614 | 0.831 | 0.967 | 0.964 | **0.993** |
| 2.5D NSCL Mao et al. (2019) disentangled | 0.897 | 0.950 | 0.964 | 0.980 | 0.986 | 0.883 | **0.968** | 0.961 | 0.969 | 0.970 |

Table 8: Accuracy of the color classifiers of the VQA models

| VQA Model | In domain test set | | | | | One shot test set | | | | |
|---|---|---|---|---|---|---|---|---|---|---|
| | Number of Training Examples | | | | | Number of Training Examples | | | | |
| | 10 | 25 | 50 | 100 | 250 | 10 | 25 | 50 | 100 | 250 |
| Our full model | 0.907 | 0.952 | 0.975 | 0.983 | 0.989 | 0.897 | **0.952** | **0.964** | **0.968** | 0.975 |
| without 3D shape prototypes | 0.917 | 0.957 | 0.620 | 0.986 | 0.987 | 0.495 | 0.445 | 0.587 | 0.967 | **0.978** |
| without shape/style disentanglement | 0.570 | 0.504 | 0.821 | 0.983 | 0.980 | 0.565 | 0.505 | 0.707 | 0.952 | 0.960 |
| without 3D shape prototypes and without shape/style disentanglement | 0.887 | 0.968 | 0.972 | 0.982 | 0.987 | 0.880 | 0.940 | 0.948 | 0.955 | 0.965 |
| Entangled disentangled features | 0.955 | 0.788 | 0.984 | 0.988 | 0.986 | **0.908** | 0.698 | 0.941 | 0.954 | 0.964 |
| InstanceNorm disentangled features + rotation-aware check | 0.798 | 0.950 | 0.963 | 0.976 | 0.986 | 0.807 | 0.888 | 0.930 | 0.955 | 0.967 |
| 2D NSCL Mao et al. (2019) | **0.970** | **0.990** | **0.993** | **0.997** | **0.997** | 0.905 | 0.872 | 0.893 | 0.948 | 0.912 |
| 2D NSCL Mao et al. (2019) without ImageNet pretraining | 0.909 | 0.960 | 0.978 | 0.982 | 0.988 | 0.778 | 0.803 | 0.808 | 0.806 | 0.842 |
| 2.5D NSCL Mao et al. (2019) | 0.877 | 0.953 | 0.967 | 0.987 | 0.993 | 0.747 | 0.786 | 0.793 | 0.828 | 0.822 |
| 2.5D NSCL Mao et al. (2019) disentangled | 0.821 | 0.954 | 0.960 | 0.977 | 0.986 | 0.709 | 0.844 | 0.805 | 0.862 | 0.738 |

Table 9: Accuracy of the material classifiers of the VQA models

| VQA Model | In domain test set | | | | | One shot test set | | | | |
|---|---|---|---|---|---|---|---|---|---|---|
| | Number of Training Examples | | | | | Number of Training Examples | | | | |
| | 10 | 25 | 50 | 100 | 250 | 10 | 25 | 50 | 100 | 250 |
| Our full model | **1.000** | **1.000** | **1.000** | **1.000** | **1.000** | **1.000** | **1.000** | **1.000** | **1.000** | **1.000** |
| without 3D shape prototypes | **1.000** | **1.000** | **1.000** | **1.000** | **1.000** | 0.522 | 0.522 | **1.000** | **1.000** | **1.000** |
| without shape/style disentanglement | **1.000** | 0.488 | **1.000** | **1.000** | **1.000** | **1.000** | 0.478 | **1.000** | **1.000** | **1.000** |
| without 3D shape prototypes and without shape/style disentanglement | **1.000** | **1.000** | **1.000** | **1.000** | **1.000** | **1.000** | **1.000** | **1.000** | **1.000** | **1.000** |
| Entangled disentangled features | 0.993 | **1.000** | **1.000** | **1.000** | **1.000** | 0.990 | **1.000** | **1.000** | **1.000** | **1.000** |
| InstanceNorm disentangled features + rotation-aware check | **1.000** | **1.000** | **1.000** | **1.000** | **1.000** | **1.000** | **1.000** | **1.000** | **1.000** | **1.000** |
| 2D NSCL Mao et al. (2019) | 0.919 | 0.974 | 0.983 | 0.991 | **1.000** | 0.755 | 0.917 | 0.867 | 0.925 | 0.919 |
| 2D NSCL Mao et al. (2019) without ImageNet pretraining | 0.872 | 0.974 | 0.993 | 0.994 | 0.999 | 0.788 | 0.963 | 0.978 | 0.966 | 0.986 |
| 2.5D NSCL Mao et al. (2019) | 0.919 | 0.965 | 0.988 | 0.994 | 0.998 | 0.869 | 0.947 | 0.974 | 0.975 | 0.941 |
| 2.5D NSCL Mao et al. (2019) disentangled | 0.627 | 0.980 | 0.972 | 0.985 | 0.982 | 0.638 | 0.978 | 0.975 | 0.983 | 0.984 |

Table 10: Accuracy of the size classifiers of the VQA models

| VQA Model | Mixed one shot test set | | | | |
|---|---|---|---|---|---|
| | Number of Training Examples | | | | |
| | 10 | 25 | 50 | 100 | 250 |
| Our full model | **0.740** | **0.804** | **0.812** | 0.820 | 0.832 |
| without 3D shape prototypes | 0.395 | 0.397 | 0.501 | 0.774 | 0.793 |
| without shape/style disentanglement | 0.448 | 0.406 | 0.601 | 0.787 | 0.777 |
| without 3D shape prototypes and without shape/style disentanglement | 0.627 | 0.724 | 0.745 | 0.757 | 0.760 |
| Entangled disentangled features | 0.611 | 0.542 | 0.807 | **0.835** | 0.838 |
| InstanceNorm disentangled features + rotation-aware check | 0.604 | 0.772 | 0.809 | 0.833 | **0.845** |
| 2D NSCL Mao et al. (2019) | 0.604 | 0.752 | 0.766 | 0.808 | 0.790 |
| 2D NSCL Mao et al. (2019) without ImageNet pretraining | 0.472 | 0.540 | 0.577 | 0.685 | 0.745 |
| 2.5D NSCL Mao et al. (2019) | 0.539 | 0.632 | 0.684 | 0.707 | 0.721 |
| 2.5D NSCL Mao et al. (2019) disentangled | 0.560 | 0.683 | 0.676 | 0.719 | 0.710 |

Table 11: VQA results on the mixed one shot dataset. This test dataset contains some objects seen in training and some are completely novel objects.

## F    QUALITATIVE RESULTS FOR ONE SHOT 3D NEURAL SCENE IMAGINATION USING LANGUAGE DESCRIPTIONS

The disentangled representations from D3DP-Nets allow us to render novel scenes which can have objects with content-style combinations not seen during training.

If the available scenes are accompanied with annotations of object categories, colors and materials, then our model can generate 3D scenes that comply with a scene description, following the method of Prabhudesai et al. (2019), assuming a parse of the scene description. We explain the experiment in detail and show some qualitative results in Section 4.4 of the main paper. We show some additional results for the same in Figure 12.

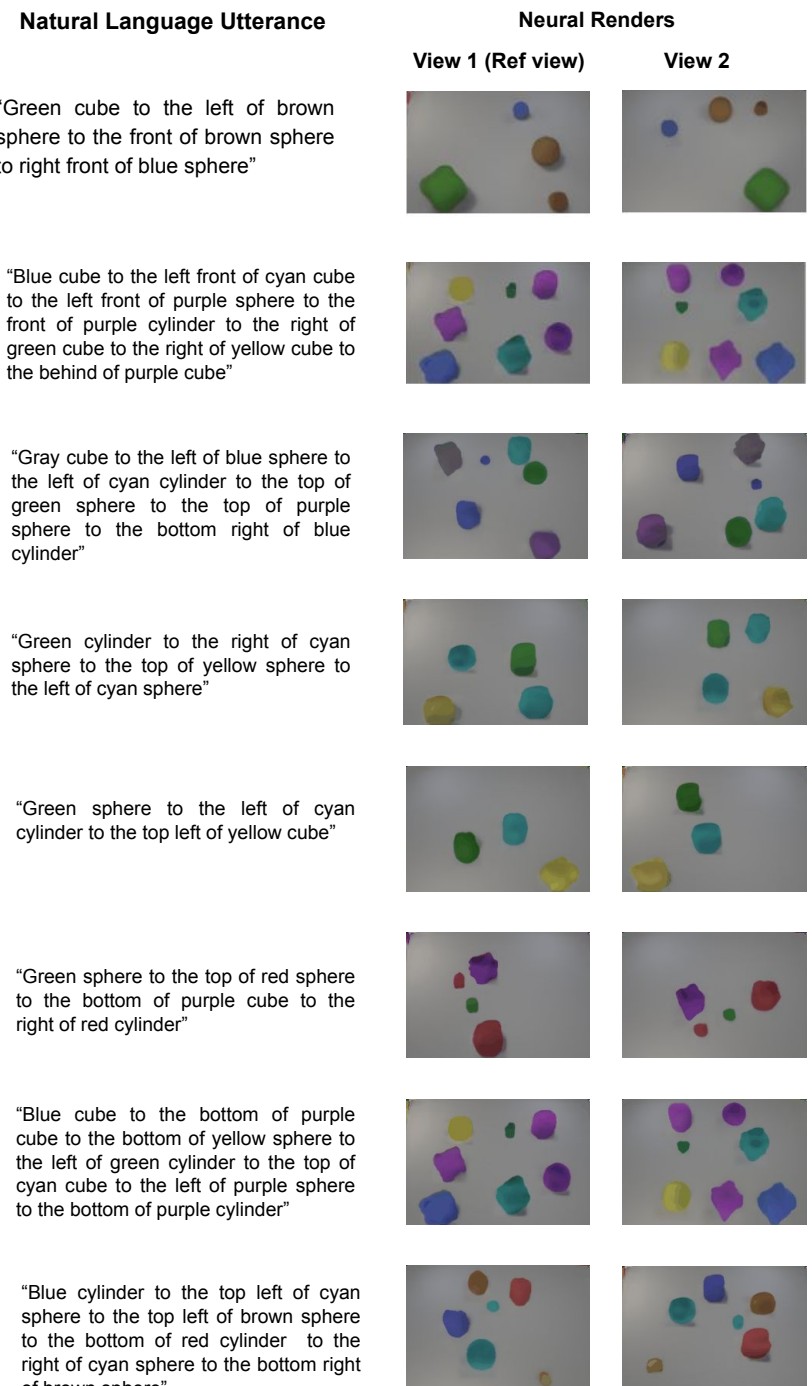

Figure 12: Generating novel scenes using only a single example for each style and content class.

