# OpenReview forum: "Disentangling 3D Prototypical Networks for Few-Shot Concept Learning"
_ICLR.cc/2021/Conference — ICLR 2021 Poster_

### Official Review · AnonReviewer2 · 2020-10-27
**Interesting submission, but experiments are not well justified**

**Rating:** 6
**Confidence:** 4

**Review:**

The paper presents a framework that combines 1) multi-view prediction for 3D reconstruction from a single image and 2) content-style disentangled representation learning using instance-norm-based auto-encoders. It aims at learning disentangled 3D representation of input images. The authors show that these methods can be applied to few-shot recognition, visual question-answering (VQA), view prediction, and image generation from texts.

The paper is well-written. It's impressive that the authors were able to build such an integrated system with so many modules. The experiments are quite extensive.

My concerns are mostly about experimental evaluation. There are some important studies missing, and other results not well-justified.

First, some important ablation studies are missing, especially in Section 4.1. The authors have used three different techniques to improve the few-shot recognition performance: 1) multi-view predictive learning, 2) content-style disentanglement, and 3) optimization-based prototypical network (the authors used gradient descent to find the rotation of the observation w.r.t. the prototype). However, the only ablation study available in Table 1 is the 3DP-Net, which uses only (1) multi-view predictive learning. Moreover, I suggest authors test the performance of other algorithms for few-shot recognition, for example, a very simple baseline algorithm from Chen (2020, https://arxiv.org/abs/2003.04390 ). Is the proposed representation algorithm only working with prototypical networks?

Second, Table 2 also misses an important baseline. Looking at the in-domain test set, we have seen that NSCL-2D has a better performance than D3DP -- the authors explained that this is because the NSCL-2D uses ImageNet pretraining. What will be the performance of NSCL without ImageNet pretraining? On the one-shot generalization test split: what will be the performance of D3DP without 3D shape primitive and without shape/style disentanglement? It seems that even with one of these two features, the proposed D3DP still has worse performance than the 2D network. It'd be great to hear from the authors if there is any intuition on the result.

I also have serious concerns about the generality and applicability of the claimed shape-style disentanglement. From the results in Table 1 (comparing D3DP and 3DP) and Table 2 (comparing D3DP w/. and w/o. style/content disentanglement), it seems that the shape-style disentanglement has shown great importance in few-shot learning settings. But, in "style features", how should materials and colors be disentangled; in "content features", how should object textures and shapes be disentangled? The limitation on generality will greatly restrict the application of this framework in real-world images.

This is also related to another minor comment about the setup of few-shot style recognition in Table 1. In CLEVR, there are 8 colors and 2 materials. So the "16 style classes" in the paper indicates that the authors are treating pairs of (color, material) as the label for objects. Such a design will not scale up wrt the type of attributes. I'd love to hear more justifications from authors. And this should be more clearly indicated in the main paper as well.

Update: I appreciate the response and have adjusted my score accordingly.

---

> ### Author Response · Authors · 2020-11-21
> **Response to review (part 1/2)**
>
> **Q1:  Ablations for 1) multi-view predictive learning, 2) content-style disentanglement, and 3) optimization-based prototypical network in Table 1.**
>
> As you have pointed out, we have included the ablation study for 1) and 2) in table 1 and table 2. In table 1, we do compare D3DP w/o. style/content disentanglement which we call as E3DP. Thanks for your suggestion on more ablation study on the prototypical networks. We replaced the nearest neighbour search of prototypes, with a linear layer which directly predicts the class probabilities, we call this network D3D. We have included results of our model without the prototypical networks In Table 1. Here are the results on the CLEVR dataset for 1-shot retrieval:
>
> |Methods|Style|Shape|
> |---|---|---|
> |D3DP|**0.61**|**0.70**|
> |D3D|0.26|0.40|
>
> As can be seen, the nearest neighbour search on prototypes plays a key role for few-shot retrieval.
>
> **Q2: Does the algorithm only work with prototypical networks?**
>
> No, our model still works without the prototypical networks, e.g., one can directly use the disentangled features to retrieve similar objects or compare attribute similarity between objects. Empirically, we found that including the prototypical networks will yield higher accuracy in few-shot concept learning since it computes the cluster mean by considering several instances as opposed to one. In addition, having explicit prototypes also allows us to add new object categories without further training.
>
> **Q3: Comparison with Chen et al. , 2020**
>
> To train the feature extractor of Chen et al, it requires many labelled “sets” of images where a set of images are labelled as “belonging to the same category.”  In contrast, our model is completely unsupervised and does not need such labelled data. We have included more comparison with this line of work in the introduction. To satisfy your request, in Table 1 we added a comparison with their Meta-Baseline model trained on Imagenet. We used the pre-trained model provided by them, in their open-source code repository. Here are the results on CLEVR dataset for 1-shot retrieval:
>
> |Methods|Style|Shape|
> |---|---|---|
> |D3DP|**0.61**|0.70|
> |Chen et al.|0.36|**0.75**|
>
> Chen et al. seem to outperform our model in shape classification since similar shape categories are present in ImageNet, but due to lack of style categories in Imagenet, it significantly underperforms in the latter.
>
> **Q4: Comparison with NSCL without Imagenet pretraining**
>
> We did include a baseline called  2.5D NSCL, which takes RGB-D image as input and does not use any Imagent pretraining. Changing inputs from RGB-D images to RGB images for 2D CNN models usually do not change much of the performance. To validate the point, we have included the results of 2D NSCL without Imagenet pretraining in Table 2. Here are the results on the In domain test set of Table 2:
>
> |Methods|10|25|50|100|250|
> |---|---|---|---|---|---|
> |Ours|**0.809**|**0.872**|**0.902**|**0.923**|**0.939**|
> |2D NSCL w/o ImgNet|0.514|0.624|0.682|0.844|0.931|
>
> As can be seen, our model significantly outperforms 2D NSCL w/o Imagenet pretraining.
>
> **Q5: D3DP without 3D shape primitive and without shape/style disentanglement in table 2?**
>
> Removing both of the modules will further degrade the performance. Thanks for this suggestion, we have added this ablation in table 2. Following are the results on the In domain test set:
>
> |Methods|10|25|50|100|250|
> |---|---|---|---|---|---|
> |Ours|**0.775**|**0.836**|**0.834**|**0.828**|**0.845**|
> |Ours w/o 3D & w/o style/shape|0.608|0.681|0.688|0.692|0.701|
>
> **Q6: Explanation for why D3DPNet ablations performed worse than 2D models in table 2.**
>
> We attribute this to the fact that we are using a limited resolution to create the 3D feature maps. One major drawback of using 3D feature maps is that these feature maps, which are indeed 4D tensors, will take a lot of GPU memory. The hardware constraint puts a limit on the resolution we can run for the 3D grids.  it’s costly for us to create a 3d grid that has the same resolution as RGB images, while 2D models can easily use high-resolution feature maps and thus result in higher performance. To address the resolution problem, we believe that integrating implicit functions with these voxel features can help us tackle this issue, similar to the Convolutional Occupancy Networks of Peng et al. We would leave this to future work.

---

> > ### Author Response · Authors · 2020-11-21
> > **Response to review (part 2/2)**
> >
> > **Q7: Can the style features be disentangled into color/material? How can the shape features be further disentangled to occupancy and texture?**
> >
> > Yes, you are correct. Currently, we are treating each combination of material/color as a unique category specifically in Table 1. However, disentangling these attributes of unsupervised should be possible. One solution is to include attribute-specific inductive biases in our network architecture, e.g., the material is a much finer style as compared to color, and thus it can be disentangled by doing AdaIn at a different spatial resolution, as suggested by the work of Karras et al. (https://arxiv.org/pdf/1812.04948.pdf). Similarly, there are biases that can be induced to disentangle shape into occupancy and texture. We will leave this for future work.
> >
> >
> >
> > Thank you for your constructive criticism, Please refer to Table 1 and 2 in the pdf, for the newly added results.

---

> > > ### Comment · AnonReviewer2 · 2020-11-25
> > > **Thanks for the update**
> > >
> > > Thanks for your response. I've raised my score to 6 given the additional results and explanations.

---

### Official Review · AnonReviewer1 · 2020-10-28

**Rating:** 6
**Confidence:** 3

**Review:**

This paper describes an approach that learns a disentangled shape and style representation of objects in a self-supervised way from RGB-D images.  The approach is based on various components, like a 3D feature volume, a bounding box detector, and a disentanglement network. Neural rendering (e.g. recomposing the various disentangled parts into an image) is used as the learning signal for self-supervision. Various applications of this representation  are shown, examples are few-shot shape learning and Visual Question Answering.

--- Strengths ---

Some applications that are enabled by the representations are very interesting. Disentangling style and shape for example allows to detect object independently of style (something easily done by humans) and generation of scenes from language utterances. Overall, it seems like a good direction to try to go for a full 3D(+style) representation to get more flexible and general models. The experiments indicate that the main concepts help in the down-stream tasks.

--- Weaknesses ---

Only synthetic samples are shown. Since the approach is fully self-supervised, it should be possible to apply it to real images. The authors show quantitative results on one, apparently, real dataset "Real veggies", but no qualitative samples are shown to give the reader an indication on how realistic this dataset is. Is it realistic? If not, what is preventing applying this approach to more realistic images and objects?

The other propose a very elaborate, modular pipeline. While this modularity does explicitely enable some of the applications that are shown, it also could be a weakness, as some modules present single points of failure which might be hard to recover from (for example when the 3D object detector fails). This could be alleviated by either showing results on realistic data and/or a controlled studies of how robust the approach is the to failures of individual parts (e.g. what happens when the detector is inaccurate?).

The paper is a bit hard to read because it is crowded. I suspect that it due to he many concepts that are introduced. The  main contribution clearly is the self-supervised learning of the representation, but a sizeable part of the paper discusses down-stream applications (which are non-trivial and are thus not discussed in sufficient detail), while other parts (for example the feature selection) would require a bit more space to be easier to understand.

--- Other ---

The example for one-shot scene generation in Figure 1 is exceptionally clean. Is this a real example that was rendered by the proposed approach?

Are you planning to release the code? An elaborate system like this would benefit strongly from this.

I suspect that the submission is severely over length. Starting from page 5, the font size is much smaller than the default font size.

Typos: page 3: "operation operation"

-- Summary ---

The proposed representation is interesting, but the huge number of applications makes the paper inaccessible. I'd argue it would be beneficial to explore and understand the representation and pipeline on a more fundamental level (i.e. can it be learned on real images? what about individual inaccuracies? when does the approach fail? what is the influence of the granularity of the 3D feature map? ....), than to jump right into elaborate higher level tasks.


--- Post rebuttal ---
I'd like to thank the authors for the response. I've updated my score in light of these additional results.

---

> ### Author Response · Authors · 2020-11-21
> **Response to review**
>
> We are encouraged that you found our applications interesting, and found this work to be in a good direction.
>
> **Q1: Model performance on realistic datasets.**
>
> The real veggie dataset is collected using a Kinect sensor, which gives us sparse depth and noisy egomotion estimate, you can find further details on the dataset in supplementary Section A. We are currently conducting our experiments on the Replica dataset of Straub et al., which contains photorealistic indoor objects. We will update the paper with these new results soon.
>
> **Q2: Robustness of the approach to the failures of individual modules.**
>
> In contrast to existing work with modular architecture where each module is learned separately, our model consists of modules that can be trained jointly end-to-end with the final objective, which allows them to adapt to a new environment. We agree that our model does depend on a good 3D detector that is trained separately from the rest of the model. In our experiments, we found our model performs comparably well when replacing the ground truth 3D boxes with detection from a  learned 3D detector. We measure the performance of our model with a learned detector,  and it achieves 0.57 and 0.62 in 1-shot style and shape retrieval tasks respectively. Model with gt object proposals performs 0.61 and 0.70. We will include similar comparisons with the Replica dataset.
>
> **Q3: More space for model and training details. Length of the paper.**
>
> Yes, you are right, the core technique novelty is to propose a self-supervised learning paradigm for learning the disentangled and 3D-aware representation. To emphasize the importance of learning such representation space, we include several down-stream applications that are critical for machine perception and concept learning. Thanks for your suggestion. We will move more details for the feature learning part to the main text. We have addressed the font size issue and the paper is within the 9-page limit.
>
> **Q4:  Are the images in figure 1 generated by the model?**
>
> No, those renders were generated from a simulator, with the intent of better explaining our tasks. You can find our neural renders in Figure 5. and supplementary videos.
>
> **Q5: Will you release your code?**
>
> Yes, we will release our code and datasets for all the applications mentioned in the paper.
>
> **Q6: Typo "operation operation"**
>
> Thanks for pointing it out. We have fixed it in the latest PDF.

---

> > ### Author Response · Authors · 2020-11-25
> > **Results on Replica dataset**
> >
> > Dear Reviewer,
> >
> > We have included the results on the Replica dataset of Straub et al (https://arxiv.org/pdf/1906.05797.pdf) in the paper.  Replica is a highly photo-realistic dataset consisting of indoor scenes of houses. You can find the results in Table 1. We also visualize the shape and style categories of Replica in Figure 4.
> >
> > | Mothod             | Style  5 shot |  Shape 5 shot | Style 1 shot | Shape 1 shot|
> > | -----------       | -----------    | -----------    | -----------    | -----------    |
> > | D3DP-Net        | **0.48**               |0.58               |**0.46**               |**0.51**               |
> > | 3DP-Net        | 0.31               |0.45               |0.27               |0.42               |
> > | 2D MUNIT     | 0.30               |**0.60**               |0.23               |0.42               |
> > | 2.5D MUNIT     | 0.23               |0.42               |0.20               |0.40               |
> > | GQN                 | 0.25               |0.31               |0.19               |0.26               |
> > | D3D-Net                 | 0.23               |0.29               |0.10               |0.14               |
> > | MB                   | 0.33               |0.32               |0.19               |0.24               |
> >
> > As can be seen, our model very well outperforms all other baselines in style/shape classification

---

### Official Review · AnonReviewer3 · 2020-10-29
**Unclear what can be really learned from the proposed method and experiments**

**Rating:** 5
**Confidence:** 3

**Review:**


The paper claims that the main contribution is "to identify the importance of using disentangled 3D feature representation for few-show learning". This is a great goal, but in my view the paper does not deliver on that front. There are several issues from my perspective

1) The model formulation aims to disentangle shape from style in a particular formulation. To achieve training of this disentanglement it requires large amounts of multi-view data of highly simplified (and in this case simulated) scenes of at most few objects at the time. I am not convinced that we will ever have such data except for highly specialized situations. Unfortunately, the paper does not discuss this major limitation and does not make any attempt to convince the reader that this is a sensible starting point for further work. The only attempt to use real data is a custom dataset called "real world veggie" that has never been used before and it's characteristics are very unclear (number of classes, number of images, number of scenes, etc - also no sample images are given)

2) In order to show the "importance of using disentangled 3D feature representation" one would expect a set of experiments that shows comparable models with and without some form of disentanglement. However, the paper does not seem to contain any such experiments as far as I understood.

3) While there are several experiments given, it is unclear how valuable the results and comparisons are. It seems that most if not all of the explored settings have not been addressed in the other works that this paper compares to. E.g. the paper [Mao et al 2019] does not seem to report any of the results given in Table 2. Similarly, the referenced papers that I checked from table 1 did not seem to report those results or this respective setting. This makes it very hard to understand if the proposed approach really has any benefit when it is used only in non-published and non-standard settings

So while the model formulation is quite sensible and combines in a meaningful way previous ideas and approaches, it remains unclear what can be taken away from this paper beyond the rather specific experiments mostly on simulated data (and some unclear real world veggie experiments)


========= post rebuttal ========

Thanks for adding the experiments for Replica - these at least seem to suggest that the approach can work on more complex scenes than shown initially in the paper.

I still think the training scheme and required data is relatively specific for this approach that it is hard to see that it will generalize beyond - but that is probably fine for a research paper.

that addresses Q1

for Q2 - please include some information in the main paper - not everyone will check the supplement and the paper should be self-contained.

I still find the comparisons and ablations weak as the particular training setup and model are the key contribution for the paper and thus without a proper ablation it is hard to know what exactly to take away. Therefore I still remain somewhat skeptical but have raised my score somewhat. To me the paper is still borderline and thus somewhere between 5 and 6 really.

---

> ### Author Response · Authors · 2020-11-21
> **Response to review**
>
> **Q1: Results on realistic and more complex scenes.**
>
> Our model is not constrained to simplified scenes with a few objects. We conduct most of our experiments using the CLEVR Dataset since it is a benchmark dataset for testing generalization on compositional concepts, as used in NSCL (Mao et al., 2019). The dataset contains cluttered scenes with up to 10 objects, which makes object detection and recognition particularly hard since many objects are heavily occluded. Our model outperforms the previous method since it learns to (1) predict the occluded part of the objects through self-supervised view prediction and (2) detect/match objects with view-invariant 3D feature representation as opposed to view-dependent 2D feature map commonly used in previous work, and thus can be less affected by the occlusions and camera views.
>
> To satisfy your request, we are currently conducting our experiments on the Replica dataset of Straub et. al., which contains photorealistic indoor scenes. We will update the paper with these new results during the rebuttal phase.
>
>
>
>
> **Q2: Details for the Real Veggie Dataset.**
>
> The Real veggie dataset contains 800 scenes of 1 to 6 vegetables placed on a table randomly. The dataset is collected using a single Kinect Camera and it contains 6 style and 7 shape classes in total. The total number of images in each scene is 6. We did include these details in Section 4.1 and Section A of supplementary.
>
>
> **Q3: Comparison between existing models with and without disentanglement.**
>
> We did include a comparison of our model with and without disentanglement in Table 1 and 2. Our model without disentanglement is identical to GRNNs of Tung et al., 2019. While your suggestion sounds reasonable, directly applying the disentanglement technique on other existing 2D-to-3D models is non-trivial since most 2D-to-3D do not operate in a feature space, and thus they cannot be trained end-to-end with the disentanglement objectives.
>
> **Q4: Experiment settings compared to previous work.**
>
> We have compared our model against SOTA baselines with similar metrics in each of the tasks. Mao et al 2019 is the SOTA in VQA for a low-label regime. While Mao et al. provided 100 labeled images for the models to adapt to new concepts, here we post a much more challenging setup where only 10 or 25 labeled images are provided for adaptation. The setup is more close to a real-world scenario where labeled data (from humans) are extremely sparse and unlabeled data (from robots) are unlimited.  In addition, we compare with Haung et al, the SOTA method in unsupervised style/content disentanglement. Although Huang et al. did not report any object retrieval/recognition accuracy in their paper, object retrieval accuracy is a common metric used in measuring the quality of different feature representations. To the best of our knowledge, these are the best models to compare against. We would be happy to compare against other competitive baselines if you have any suggestions on this.

---

> > ### Author Response · Authors · 2020-11-25
> > **Results on Replica dataset**
> >
> > Dear Reviewer,
> >
> > We have included the results on the Replica dataset of Straub et al (https://arxiv.org/pdf/1906.05797.pdf) in the paper.  Replica is a highly photo-realistic dataset consisting of indoor scenes of houses. You can find the results in Table 1. We also visualize the shape and style categories of Replica in Figure 4.
> >
> > | Mothod             | Style  5 shot |  Shape 5 shot | Style 1 shot | Shape 1 shot|
> > | -----------       | -----------    | -----------    | -----------    | -----------    |
> > | D3DP-Net        | **0.48**               |0.58               |**0.46**               |**0.51**               |
> > | 3DP-Net        | 0.31               |0.45               |0.27               |0.42               |
> > | 2D MUNIT     | 0.30               |**0.60**               |0.23               |0.42               |
> > | 2.5D MUNIT     | 0.23               |0.42               |0.20               |0.40               |
> > | GQN                 | 0.25               |0.31               |0.19               |0.26               |
> > | D3D-Net                 | 0.23               |0.29               |0.10               |0.14               |
> > | MB                   | 0.33               |0.32               |0.19               |0.24               |
> >
> > As can be seen, our model very well outperforms all other baselines in style/shape classification

---

### Official Review · AnonReviewer4 · 2020-10-30
**Well-designed neural architecture for few-shot concept learning with state-of-the-art results**

**Rating:** 7
**Confidence:** 4

**Review:**

This paper presents a modular network architecture for few-shot concept learning. The architecture consists of image-to-scene module that maps input RGBD images to 3D scene features and an object-centric disentangling auto-encoder that crops object features to generate shape and style codes, and finally a neural rendering module that put back the reconstructed object and background features to image domain. The proposed network is verified in few-short recognition task and VQA task with comparisons to state-of-the-art methods.

+This paper is well-written and the core ideas and network design are well illustrated and explained in the paper and supp video. This paper also presents comprehensive experiments including sufficient details in supp to support the claims, which also makes the paper more reproducible.
+State-of-the-art results on few-shot recognition, especially on one-shot recognition. The object-centric disentangling and 3D shape prototype learning seem to play complementary roles for the results.

-The success of the proposed method seems to be limited to the data being considered in this paper, rigid 3D objects with different appearances. It is not clear how the proposed methods, e.g. AdaIn disentangling and rotation invariant prototype can be easily generalized to other scenarios including deformation (pointed out in the paper) and part-based composition.

---

> ### Author Response · Authors · 2020-11-21
> **Response to review**
>
> We are encouraged that you found the paper to be “well-written,” “presenting comprehensive experiments with sufficient details, and “achieving state-of-the-art results on few-shot recognition.” Below we address your concerns.
>
> **Q1: Handling non-rigid objects.**
>
> You are right that the presenting model does not support concept learning for deformable objects. But this is not a restriction to the proposed learning framework. To handle deformed objects, one approach is to simply use a pooling operation to pool the 3D prototypes to 1D prototypes and conduct the matching in this 1D representation space. The major drawback of this approach is that the model can not infer object pose by rotation check since the representation already loses its 3 spatial dimensions. A better approach would be to learn 3D part or 3D keypoint detectors as opposed to whole object detectors, and then quantize and correspond the parts using our approach. We leave this to future work.

---

### Author Response · Authors · 2020-11-25
**General Response**

We thank all the reviewers for their detailed feedback and useful suggestions! We have tried our best to address all the concerns of the reviewers in the individual responses. We have also updated our manuscript with the requested experiments. We summarize the major changes below:

1. Added results for few-shot shape/style classification (Table 1 and Figure 4) on a highly photo-realistic Replica dataset.
2. Introduced a new supervised baseline of Chen et al in Table 1.
3. Included an ablation study for Prototypical Network in Table 1.
4. Updated Table 2 with results of 2D NSCL without Imagenet pretraining:
5. Added D3DP model without 3D shape primitive and without shape/style disentanglement in Table 2.
3. Included Replica dataset collection details in Appendix A.
7. Fixed the font issue and spelling mistakes.

---

### Decision · Program_Chairs · 2021-01-07
**Final Decision**

**Decision:**

Accept (Poster)

**Comment:**

This meta-review is written after considering the reviews, the authors’ responses, the discussion, and the paper itself.

The paper proposes a system for learning disentangled object-centric 3D-based representations of scenes and shows that the proposed model works well on several tasks, including few-shot classification and VQA.

The reviewers point out that the direction is important (R1, R3), the model is sensible (R2), and the reported results are good (R1, R4); on the downside, they note that the system is complicated (R1), the considered datasets are relatively simplistic (R1, R3, R4), some ablations are missing (R2, R3), and comparisons with baselines are not necessarily convincing (R2, R3). The authors did a good job of addressing the concerns in the rebuttal, by reporting additional ablation results, baselines, and experiments on the realistic Replica dataset.

All in all, I recommend acceptance. The direction of the work is important and complex, the experimental evaluation presented in the paper is extensive, and the results are good relative to relevant baselines. On the downside, the proposed system and the paper are somewhat complicated and overwhelming, which may limit the benefit for the readers. I hope the authors will take this into account in the future.